# Simulating the Early Holocene demise of the Laurentide Ice Sheet with BISICLES (public trunk revision 3298)

Ilkka S.O. Matero[1,2], Lauren J. Gregoire[1], and Ruza F. Ivanovic[1]

[1]University of Leeds, United Kingdom
[2]Alfred Wegener Institute, Germany

**Correspondence:** Ilkka Matero (ilkka.matero@awi.de)

**Abstract.** Simulating the demise of the Laurentide Ice Sheet covering the Hudson Bay in the early Holocene (10-7 ka) is important for understanding the role of accelerated changes in ice sheet topography and melt in the '8.2 ka event', a century long cooling of the Northern Hemisphere by several degrees. Freshwater released from the ice sheet through a surface mass balance instability (known as the *saddle collapse*) has been suggested as a major forcing for the 8.2 ka event, but the temporal

evolution of this pulse has not been constrained. Dynamical ice loss and marine interactions could have significantly accelerated the ice sheet demise, but simulating such processes requires computationally expensive models that are difficult to configure and are often impractical for simulating past ice sheets. Here, we developed an ice sheet model setup for studying the Laurentide Ice Sheet's Hudson Bay saddle collapse and the associated meltwater pulse in unprecedented detail using the BISICLES ice sheet model, an efficient marine ice sheet model of the latest generation, capable of refinement to kilometre-scale resolution

and higher-order ice flow physics. The setup draws on previous efforts to model the deglaciation of the North American Ice Sheet for initialising the ice sheet temperature, recent ice sheet reconstructions for developing the topography of the region and ice sheet, and output from a general circulation model for a representation of the climatic forcing. The modelled deglaciation is in agreement with the reconstructed extent of the ice sheet and the associated meltwater pulse has realistic timing. Furthermore, the peak magnitude of the modelled meltwater equivalent (0.07-0.13 Sv) is compatible with geological estimates of freshwater

discharge through the Hudson Strait. The results demonstrate that while improved representation of the glacial dynamics and marine interactions are key for correctly simulating the pattern of early Holocene ice sheet retreat, surface mass balance introduces by far the most uncertainty. The new model configuration presented here provides future opportunities to quantify the range of plausible amplitudes and durations of a Hudson Bay ice saddle collapse meltwater pulse and its role in forcing the 8.2 ka event.

*Copyright statement.* TEXT

# 1 Introduction

A centennial-scale meltwater pulse produced by the collapse of an ice sheet that once covered Hudson Bay has been shown to be the possible driver of the most pronounced climatic perturbation of the Holocene, the 8.2 ka event (Matero et al., 2017), supported by recent geochemical evidence (Lochte et al., 2019). This so-called Hudson Bay Saddle Collapse resulted from an acceleration in melting as the region connecting the Keewatin and Labrador ice domes became subject to increasingly negative surface mass balance (SMB) through a positive feedback with surface lowering (Gregoire et al., 2012). Gregoire et al. (2012) simulated an acceleration of melt lasting about 800 years and peaking at 0.2 Sv, which produced a 2.5 m contribution to sea level rise in 200 years caused by the separation of three ice sheet domes around Hudson Bay. While the saddle collapse mechanism described in this study is robust, the detailed evolution of the ice sheet did not fully match empirical reconstructions, with the separation of the Labrador, Keewatin and Baffin domes not occurring in the order suggested by geological evidence (Dyke, 2004). This mismatch between model results and geological evidence could have impacted the duration and amplitude of the simulated meltwater pulse. In addition, the deglaciation of the ice saddle could have been further accelerated through dynamic ice sheet instabilities such as grounding line destabilisation, ice streaming and increased calving at the marine terminus (Bassis et al., 2017). Contemporary continental ice sheets with marine margins are highly sensitive to these localised features and require up to kilometre-scale resolution to resolve them properly (Durand et al., 2009; Cornford et al., 2016). High-resolution ice sheet modelling simulations of these processes have not been previously conducted for the early-Holocene Laurentide Ice Sheet (LIS), and as a result a detailed model representation of the LIS saddle collapse and the resulting meltwater pulse remain knowledge gaps to be filled.

It is important to constrain the evolution of the saddle collapse in order to better understand the major forcing of the 8.2 ka event, as both the modelled regional climate responses and the ocean circulation are sensitive to the duration and magnitude of the meltwater pulse (Matero et al., 2017). Changes in topography may play a secondary role, for example by modifying atmospheric circulation and thereby influencing North Atlantic Gyre circulation, but in this instance they are likely to be a much weaker forcing for climate change than freshwater released by the melting ice (Gregoire et al., 2018). Accurately representing the dynamical processes at marine margins of ice sheets has previously been challenging for studies of the deglaciation of continental-scale ice sheets, which takes place over several millennia, due to the high computational cost of the models. Application of Adaptive Mesh Refinement (AMR) is a pragmatic approach towards solving this problem, operating at a fine resolution in the dynamic regions, while keeping a coarser resolution where the ice is quiescent. AMR has been applied for simulating contemporary Antarctic and Greenland Ice Sheets in high detail using the BISICLES ice sheet model (Cornford et al., 2013, 2015; Favier et al., 2014; Lee et al., 2015; Nias et al., 2016), and through good fit with observed deglaciation and migration of the grounding line, these studies have demonstrated the usefulness of the model. They showed that the movement of the grounding line, overall mass loss and evolution of fast-flowing ice streams are all sensitive to increasing the mesh resolution and features in bed geometry (Cornford et al., 2015; Lee et al., 2015). A recent study applied BISICLES in a palaeo setting, focussing on simulating part of the British and Irish Ice Sheet, and highlighted the importance of marine interactions and marine ice sheet instability during the last deglaciation of the British Isles (Gandy et al., 2018). However, applications of

the BISICLES ice sheet model to simulating past ice sheet evolution have so far been limited to small ice sheets and have used idealised climate forcings. The present study addresses this directly, developing the model setup required to apply the BISICLES ice sheet model to simulating the demise of the Laurentide Ice Sheet in the early Holocene, providing a useful model for (a) understanding the late history of this ice sheet and (b) improving constraints on meltwater, surface energy balance and
topographic forcing of climate around this time.

Ice sheets experience mass fluctuations as a result of the interplay between snow accumulation and ablation at the surfaces of the ice sheet and dynamic ice loss through calving (Bauer and Ganopolski, 2017). Representing the modes of ice loss in palaeo settings is challenging due to a lack of observational data for constraining the SMB as well as ice flow. The evolution of a simulated palaeo ice sheet cannot be directly compared with observations of SMB or ice flow, but instead reconstructions of
the evolution LIS extent and the positions of ice streams provide end-member constraints with which to compare and evaluate the results of the simulations. Two of the most recent LIS reconstructions are the commonly used ICE-6G_c (VM5a) reconstruction (henceforth "ICE-6G_c" Peltier et al., 2015) and the North American component of the GLAC-1D reconstruction (Tarasov et al., 2012). The ICE-6G_c reconstruction is based on Glacial Isostatic Adjustment (GIA) modelling and constrained by GPS observations of vertical motion of the crust, ice margin chronologies (Dyke, 2004), sea level rise (SLR) data and
space-based gravimetric measurements (Peltier et al., 2015). The global GLAC-1D reconstruction is combined from multiple sources (described in Ivanovic et al., 2016), of which the North American component is based on a dynamical ice sheet model constrained with relative sea level data (Tarasov et al., 2012, and references therein) and ice sheet extent from Dyke (2004). The evolution of the extent of LIS over the early-Holocene has been reconstructed from the timing of when individual locations became ice-free with an estimated error of ~500–800 years (Dyke, 2004; Margold et al., 2018). In terms of ice volume, the
reconstructions estimate that in the early Holocene (10 ka), LIS held ~11–18 m (GLAC-1D) and ~12.5 m (ICE-6G_c) global mean sea level rise equivalent, having already lost more than 80% of its Last Glacial Maximum mass.

We built a configuration of the BISICLES ice sheet model (Cornford et al., 2013) to simulate the early-Holocene LIS deglaciation based on information from the reconstructed evolution of the ice sheet (ICE-6G_c and GLAC-1D) and a previous experiment that encompassed the deglaciation (Gregoire et al., 2012). The model setup was developed with the ultimate aim
of improving knowledge of the Hudson Bay saddle collapse through simulating the deglaciation with an updated climate forcing, finer spatial resolution and a more sophisticated representation of dynamic ice flow (including ice streams) compared to previous modelling studies of the time period (Marshall et al., 2002; Tarasov et al., 2012; Gregoire et al., 2012). As part of the setup, a selection of key parameters are varied individually to assess their role in model performance. The key parameters affecting ice flow are basal traction and the internal temperature structure of the ice sheet, and the key parameters affecting
SMB are the Positive-Degree-Day (PDD) factors defining the amount of snow and ice melt in response to the climate forcing (the SMB is represented through a PDD scheme, Rutt et al., 2009). In addition to these parameters and the mesh resolution, the effect of scaling the precipitation field is also evaluated as part of the sensitivity study, because the palaeo precipitation field from a single general circulation model (GCM) can exhibit significant regional biases (Knutti et al., 2010; Braconnot et al., 2012). This technical manuscript presents the model configuration and its sensitivity to the aforementioned model parameters.
Understanding the sensitivity of the results to choices in modelling setup for such novel, palaeo applications is an important

technical step for eventually being able to reach a better understanding of the recorded events through further modelling work; for example, to quantify the range of plausible rates of meltwater fluxes, which would then provide a powerful (hitherto elusive) test for climate model performance (Schmidt and LeGrande, 2005).

## 2   The BISICLES ice sheet model

BISICLES is a vertically integrated ice flow model based on the 'L1L2' dynamical scheme devised by Schoof and Hindmarsh (2010), described in detail by Cornford et al. (2013). It has 10 vertical layers, which increase in thickness from 2% of ice thickness near the base to 15% of ice thickness near the surface. Ice in the model is assumed to be in hydrostatic equilibrium, with the weight of the ice mass being balanced by a pressure gradient at the lower surface. Sufficient changes in thickness at or near the grounding line thus define the movement of the grounding line as a result of transition between grounded and floating
ice. This movement of grounding line is an important feature in determining the evolution of marine ice sheets, and has been shown to require up to kilometre-scale resolution to resolve adequately (Favier et al., 2014; Cornford et al., 2016).

Numerical modelling of entire continental-scale ice sheets with kilometre-scale resolution and higher-order physics is currently unfeasible due to the high computational cost required. Therefore, to address the need for high resolution in specific parts of the ice sheet while simultaneously limiting the computational cost, BISICLES includes a block-structured adaptive mesh
refinement (AMR) method. Using the AMR, the model automatically refines, maintains or coarsens the horizontal resolution in regions as necessary. Resolution can be refined based on stress-balance equations and adjacency to the grounding line and shear margins (Cornford et al., 2013; Favier et al., 2014). The computational efficiency of the model is further enhanced with the capability for MPI-based parallel computing.

The ice thickness $h$ and horizontal velocity vector $\boldsymbol{u}$ satisfy a mass conservation equation for vertically integrated transport
of incompressible material, which can be expressed as:

$$\frac{\delta h}{\delta t} = - \bigtriangledown \cdot [\mathbf{u}h] + M_s - M_b, \tag{1}$$

in which $M_s$ is the surface mass balance rate and $M_b$ is the basal melting rate and $\bigtriangledown \cdot [\mathbf{u}h]$ represents horizontal advection of mass.

### 2.1   Ice flow

The model describes an ice mass evolving through three-dimensional, shear-thinning flow driven by gravity using a depth-integrated ice flow model devised by Schoof and Hindmarsh (2010). The deviatoric stresses and the strain rates are related through a stress balance equation:

$$\bigtriangledown \cdot [\varphi h \mu (2\dot{\epsilon} + 2tr(\dot{\epsilon})\mathbf{I}) + \boldsymbol{\tau}_b = \rho_i gh \bigtriangledown s, \tag{2}$$

in which $\dot{\epsilon}$ is the horizontal strain rate tensor and $\boldsymbol{I}$ is the identity tensor. $\tau_b$ is the basal traction (see sections 2.2 and 3.3).
$\varphi h \mu$ is the vertically-integrated effective viscosity, and is calculated from the vertically varying effective viscosity $\mu$ derived

from Glen's flow law, and the stiffening factor $\varphi$ (Cornford et al., 2015; Nias et al., 2016). The vertically-integrated effective viscosity varies spatially, mainly depending on whether the ice is grounded or not, the basal traction, the ice temperature and how fractured the ice is.

The stiffening factor coefficient $\varphi$ accounts for variations in several factors including temperature and minor fractures in the ice. For contemporary ice sheets, the stiffening factor is typically solved based on surface ice velocities using inverse methods and is used in the model as an additional tuning parameter when calibrating ice flow to match observations (e.g. Cornford et al., 2015). Due to the lack of direct observations of ice surface velocities for the early-Holocene LIS. We use the default value of $\varphi = 1$ for the entire ice sheet. Given the standard Glen's flow law exponent of $n = 3$, $\mu$ satisfies

$$2\mu A(T)(4\mu^2 \dot{\epsilon}^2 + |\rho_i g(s-z) \bigtriangledown s|^2) = 1, \tag{3}$$

where $z$ is the depth and $A(T)$ is a rate factor that depends on the ice temperature $T$ through the Arrhenius law (Hooke, 1981).

## 2.2 Basal processes

Boundary conditions at the base of the ice vary between different parts of the ice sheet. Where the ice is floating, there is no basal traction and the normal stress at the base matches the hydrostatic water pressure. Elsewhere the ice is in contact with either bedrock or glacial sediments, of which neither allow flow normal to the base. Following equation 2, the basal traction $\tau_b$ is a major controlling factor of the ice flow speed in the model. For ice in contact with bedrock, it can be expressed as:

$$\boldsymbol{\tau}_b = -C|\mu|^{m-1}\boldsymbol{u}, \tag{4}$$

with $h(\frac{\rho_i}{\rho_w}) > -r$ as the flotation criteria, where $r$ is bedrock elevation. $C$ is the basal traction coefficient, which can be set to a spatially varying field and used as a tunable model parameter. $\tau_b$ is assumed to satisfy either a non-linear power law, where $m = 1/3$, or a linear viscous relation, where $m = 1$ (with $m = 1$ used in this study). $C$ is typically solved based on the surface ice flow speed using inverse methods for contemporary ice sheets, but for the study presented here, a parametrised field is used based on the abundance and thickness of glacial sediments and bedrock elevation (see section 3.3).

Basal heat flux only affects the temperature of the ice in the model, which can impact the ice flow by changing the effective viscosity $\mu$ (eq. 3). This is a simplification of the subglacial hydrology, which comprises several processes that can alter the dynamics of ice flow on different time scales (Clarke, 2005; Gladstone et al., 2014). The motion of the base of an ice sheet can be due to (typically plastic) deformation of the underlying sediment or the base of the ice sheet actually sliding on top of the underlying substrate (Gladstone et al., 2014). The sliding is facilitated by the presence and distribution of liquid water at the base, and the yield stress of the deformation of the till is strongly dependent on effective pressure (Iverson, 2010). The conditions at the ice-bed interface (melting or non-melting), availability of liquid water and mechanical properties of the till (soft or hard) thus control the processes that can be activated. A simple hydrology scheme has recently been added to BISICLES

to allow for the self generation of ice streams by representing basal sliding with a coulomb law sensitive to the presence of till water [Gandy et al.]. However, this scheme not included in the version of BISICLES used here. Instead, we capture the effect of these processes by imposing a spatially variable map of the basal traction coefficient $C$.

Basal traction does not impact the flow of floating ice. However, the ice shelves can have a buttressing effect on the ice flow upstream of the grounding line if the shelves are laterally bounded (e.g. Dupont and Alley, 2005). The melting rate under floating ice sheets can thus have a major impact on the flow of ice streams by influencing the buttressing effect (Schoof, 2007). The contemporary Antarctic floating glaciers have been estimated to undergo melt rates in the range of 0–43 ma$^{-1}$ (Rignot and Jacobs, 2002; Rignot et al., 2013). The sensitivity of the modelled ice sheet and the Hudson Strait ice stream to varying rates for this parameter is discussed in section 4.6.

## 2.3 Calving

BISICLES can represent calving in different ways. Here, we use the crevasse calving model of Taylor (2016), which defines the calving front as being where surface or basal crevasses result in a full-thickness fracturing of the terminus. The model calculates the depth of crevasse penetration for the entire domain at both grounded and floating termini. The equations that calculate the penetration depth of both surface and basal crevasses were developed based on earlier studies on calving of tidewater glaciers and marine outlets (Benn et al., 2007a; Nick et al., 2010). A full description of the BISICLES Benn calving model is available in the PhD Thesis of Taylor (2016).

## 2.4 Surface mass balance and climate inputs

We simulate the surface mass balance of the Laurentide Ice Sheet with a PDD surface mass balance model (Rutt et al., 2009) driven by climatological means from a GCM following a similar methodology to (Gregoire et al., 2012, 2015, 2016), but with the following differences. We drive the PDD model with monthly temperature and precipitation from a series of HadCM3 GCM palaeoclimate 'snapshots' (i.e. equilibrium simulations) run at 1-ka intervals for 26-21 ka and 500-year intervals for 21-0 ka. They are the same simulations used and described in more detail by Morris et al. (2018), Swindles et al. (2018) and Gandy et al. (2018). The snapshots represent a refinement from earlier HadCM3 simulations (Singarayer et al., 2011; Singarayer and Valdes, 2010), and have been updated according to boundary conditions provided for the Palaeoclimate Model Intercomparison Project Phase 4 protocol for simulations of the last deglaciation (version 1; Ivanovic et al., 2016), using the ICE-6G_c reconstruction (Peltier et al., 2015).

For the purpose of this work, we scaled down the climate model output to a resolution 0.5 × 0.5 degrees using a bivariate spline interpolation method. The downscaled resolution is approximately equal to 50 × 50 km at mid-latitudes. To generate a transient climate forcing, we linearly interpolate between the climate means calculated from the last 50 years of each 500-year snapshot performed for the early Holocene.

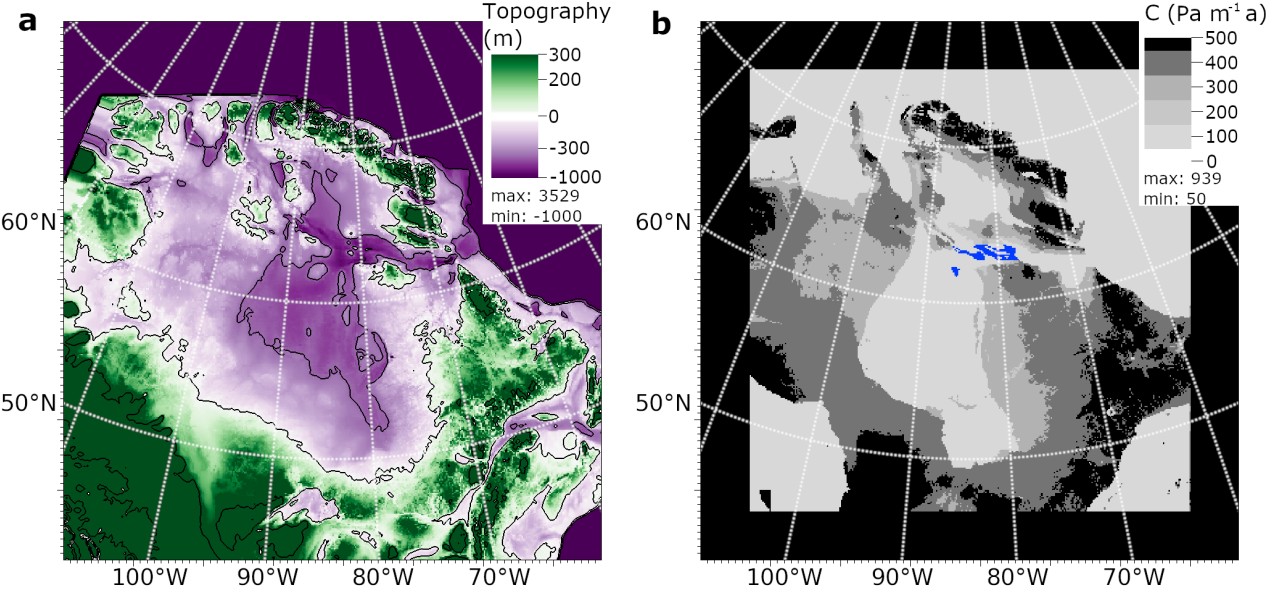

**Figure 1. (a)** The 'ETOPO 10 ka' topography map centred at Hudson Bay on a 10 km grid. The basal topography in the northern part of the grid (Greenland and Canadian archipelago) is set to -1000 m to avoid ice sheet formation in these regions. The black lines show contours every 400 m. **(b)** Basal traction coefficient map used in the 'standard' simulation. The basal traction coefficient (C) values are calculated based on bedrock elevation and sediment coverage. The domain was divided into three different types of sediment coverage: bare bedrock, sediment-covered and submerged bare bedrock in the Hudson Strait (highlighted in blue, $C = 80\ Pa\ m^{-1}\ a$). The outermost 35 grid cells along both axes are initialised with a high basal traction coefficient ($C = 450\ Pa\ m^{-1}\ a$) to avoid ice flow reaching the edges of the domain.

## 3 Model setup and experimental design

By the early Holocene, the LIS has significantly retreated from its Last Glacial Maximum position and is far from being at equilibrium with the climate. Thus, we chose to initialise the geometry of the ice sheet, its thickness and the bedrock topography based on an available ice sheet reconstruction. We used the ICE-6G_c reconstruction, which is designed to be consistent with geological records of ice sheet extent, sea level and measurements of modern rates of isostatic adjustment (Peltier et al., 2015). However, ice thickness in this reconstruction is known to be inconsistent with fundamental ice sheet dynamics (Stuhne and Peltier, 2017) and is also likely inconsistent with the early Holocene climate simulated by HadCM3 because the climate and ice sheet were not generated in fully (i.e. two-way) coupled framework (e.g. HadCM3 used ICE-6G_c as a boundary condition, but the ice sheet was not able to subsequently evolve depending on the resulting climate). Thus, to allow the ice sheet geometry and velocity to adjust to the model physics and climate forcing, we start our simulations with a spin-up phase from 10 ka to 9.0 ka and continue the simulations over our period of interest 9 - 8 ka during which the separation of the Labrador and Keewatin ice domes occurs (Ullman et al., 2016).

Several parameters are needed to initialise the model, and they are generally poorly constrained by observations for palaeo ice sheets. Ice thickness data $h_o(x,y)$, together with surface mass balance $M_s(t,x,y)$ and basal melt rates $M_b(t,x,y)$ are required to initialise the model for solving the equation for conservation of mass (eq. 1). These, together with a map of bedrock elevation $b(x,y)$, ice temperature field $T(x,y,z)$, basal traction coefficient $C(x,y)$ and an ice stiffening factor $\Phi(x,y)$ allow for solving the

equation of ice flow (eq. 2). The methods used to produce the initial conditions are described in the following subsections, and the ice velocity is initially 0 m a$^{-1}$.

## 3.1    Model domain and basal topography

The model domain at the coarsest level $\Omega^0$ consists of a grid of $384 \times 384$ rectangular cells of $10 \times 10$ km horizontal resolution centred on the Hudson Bay (fig. 1). The projection used is the Lambert Azimuthal Equal Area (LAEA) with a point of origin

at 45° N, 95° W, and false easting and northing of 1648.38 km and 202.32 km respectively. The resolution is comparable to what is typically applied in long-term simulations of the modern-day Greenland Ice Sheet using regular equidistant grids (5-20 km; Goelzer et al., 2017), and what has been used as the base resolution in earlier continental-scale ice sheet simulations using BISICLES (8 km for the Antarctic Ice Sheet in Cornford et al., 2016; 4 km for Greenland in Lee et al., 2015). This base resolution was chosen after initial test runs (not shown here), in which doubling the base resolution to 5 km did not result in

significant changes in the ice sheet geometry when keeping the finest resolution allowed in the simulation at 5 km, but resulted in a three-fold increase in the computation cost with the early model setup. The maximum resolution that it is feasible to run the model over this domain for the duration of 2000 years with the current setup is $\Omega^2$ = 2.5 km. An average run speed of 396 model years per day is reached with 96 processors on Tier 3 (ARC3, University of Leeds) high performance computing facilities, whereas test runs with one more level of refinement $\Omega^3$ = 1.25 km slowed the initial run speed to less than 10 model

years per day.

To produce the bedrock elevation $b(x,y)$, a rectangular bivariate spline interpolation method is used to combine a high-resolution (1 arc-minute resolution) present-day basal topography relief model (Amante and Eakins, 2009) with the difference between the 10 ka and present-day basal topographies from the ICE-6G_c reconstruction. Initial tests were run using the basal topography from the 10 ka time slice of the ICE-6G_c reconstruction, which quickly highlighted the need for finer resolution than the native horizontal grid of the reconstruction (1° × 1°). This was because the blocky structure of the base in ICE-

6G_c resulted in unrealistically steep vertical gradients in the surface elevation $s(t,x,y)$. The resulting high-resolution basal topography map is resampled from the swath dataset and projected onto the model grid at using a nearest neighbour method to produce the 'ETOPO 10 ka' topography map used for initialising the ice sheet model (fig. 1a). The topography was generated at the coarsest resolution of the model setup and is upscaled as part of the adaptive mesh refinement. When a part of the model

domain is refined, the grid cells at the base of this region are linearly interpolated to match the level of refinement of the overlying ice sheet. The basal topography for the regions of Canadian archipelago and Greenland included in the domain are set to -1000 m to avoid ice sheet build-up in these regions. These areas lack relevance to this study and were excluded in order to save computational cost by avoiding computing the ice velocities there.

The transient isostatic rebound of the crust is not included in the simulations as this feature was not available when this work was carried out. The maximum crustal uplift between 10 ka and the present day in the region is of the order of ~500 m in the ICE-6G_c reconstruction. Therefore, not including isostatic rebound in our simulations could have resulted in a minor overestimation of the simulated surface temperatures in some regions due to the temperature lapse rate. This effect is however

estimated to be small (< 1 K) because the majority of the post-glacial rebound happening after the LIS deglaciation (Peltier et al., 2015).

## 3.2 Ice thickness and temperature

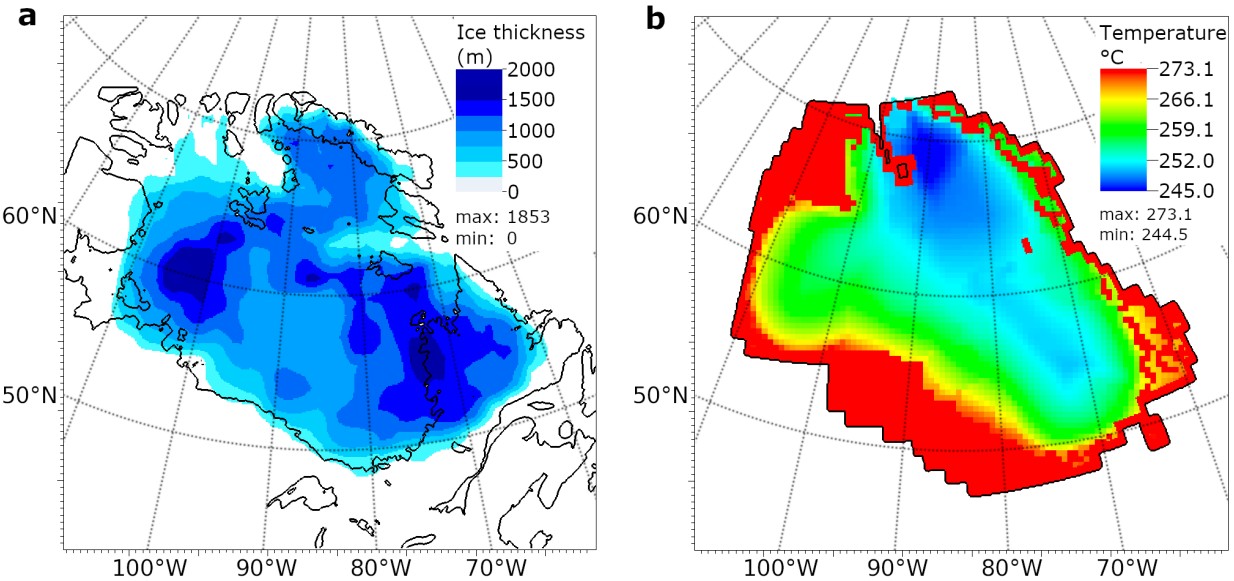

**Figure 2. (a)** Initial ice thickness (based on ICE-6G_c 10 ka time slice; Peltier et al., 2015), with the coastline shown in black. **(b)** Initial ice temperature at the lowest level of the ice sheet (based on 9 ka time slice of the simulation by Gregoire et al., 2012).

The thickness of the ice sheet (fig. 2a) is initialised from the 10 ka time slice of the ICE-6G_c reconstruction using a similar approach to that used for generating the initial topography. The thickness data is first interpolated from low to high resolution

(from $1°$ to 1 arc-minute resolution) using a rectangular bivariate spline interpolation method, and then resampled to the model grid using a nearest neighbour method.

The volume and area of the LIS in the ICE-6G_c at 10 ka do not conform to the empirical relationship of volume to area ratios of contemporary ice sheets (fig. 4 in Ullman et al., 2016; Paterson, 2016). According to this ratio, the volume of the ice sheet would have been ~$11.7 \cdot 10^6$ km$^3$ $\pm 12\%$ based on the reconstructed 10 ka area in ICE-6G_c of $6.08 \cdot 10^6$ km$^2$ (Peltier

et al., 2015). This volume would be equivalent to ~29.6 m of eustatic SLR, which is higher than the volumetric estimate of LIS at 10 ka equivalent to ~9.3 m of eustatic SLR in ICE-6G_c 10 ka time slice (Peltier et al., 2015). It is uncertain if a volume to

area ratio based on contemporary stable ice sheets and glaciers is directly applicable for a deglaciating ice sheet, but the large discrepancy between the volumetric estimates suggests that the ice volume in ICE-6G_c at 10 ka is likely underestimated as the LIS area is better constrained than its volume for this time period.

The initial temperature (fig. 2b) of the ice sheet is based on the 9 ka time-slice of a previous LIS deglaciation simulation using the GLIMMER-CISM ice sheet model (Gregoire et al., 2012). The original data is on a LAEA grid with a common point of origin with our experiment (45° N, 95° W), but with a larger domain covering most of North America (the GLIMMER-CISM study examined the evolution of the whole North American Ice Sheet over the last glacial cycle). The data is interpolated to the smaller domain and higher resolution using the rectangular bivariate spline interpolation method. The 9 ka time slice of the previous study was chosen because it presents a close fit to the extent of the ICE-6G_c ice sheet at 10 ka, which is the starting point of the transient simulations presented here. The two do not match exactly, and for grid cells where ICE-6G_c indicates that ice should be present but the GLIMMER-CISM simulation is ice-free, the temperature is initialised to 0 °C for all vertical levels to rectify the discrepancy. Due to using the PDD model (section 2.4) for modelling the surface mass balance, this does not impact the ablation rate directly in the areas, but encourages melting due to promoting ice flow (eq. 2) of the extensive thin areas of the initial ice sheet.

## 3.3   Basal traction

Basal traction is a major factor determining the ice flow in BISICLES (eq. 2), and one of several factors that have been identified as potential controls for ice flow and streaming locations in ice sheets (Winsborrow et al., 2010; Stokes et al., 2016). The presence of water and thickness of subglacial sediments have an influence on the basal traction, as till deformation can accelerate basal flow. This has been shown to be important for the ice dynamics in earlier modelling studies of the LIS (Marshall et al., 2002; Tarasov and Peltier, 2004) and is key to enable ice stream to self-generate (Payne and Dongelmans, 1997). Recently, A simple basal sliding scheme has been added into BISICLES. The scheme uses a Coulomb sliding law sensitive to the presence of till water and is able to simulate well the ice streams of the British and Irish Ice Sheet (Gandy et al., 2019). However, the version of BISICLES we used here does not yet include these processes of basal hydrology that are needed to generate ice streams. We thus adjust the basal traction coefficient $C(x,y)$ to account for these effects as is typically done in simulations of modern ice sheets. To represent this dependency of basal traction on subglacial properties and to determine the basal traction coefficient $C(x,y)$, the domain was divided into three different types of grid cells according to sediment coverage with the division based on a geological map of North America by Reed et al. (2005).

In addition to the underlying substrate, Basal topography and topographic troughs are a major factor influencing the generation and positioning of ice streams (Paterson, 2016; Winsborrow et al., 2010). This effect was incorporated by relating the basal traction coefficient $C$ for the regions of bare bedrock as a function of bedrock elevation $b$ using:

$$C(x,y) = b(x,y)C_i + C_{sl}, \tag{5}$$

where $C_i$ is the increment of the basal traction coefficient with elevation (0.3 m$^{-1}$ in the 'standard' simulation), and $C_{sl}$ is the value for bedrock at sea level ($C = 400$ Pa m$^{-1}$ a in 'standard').

A fixed value is used in areas with a sediment coverage (fig. 1b). The range of $C$ -values chosen for the initialisation was based on values from modern-day simulations of the West Antarctic Ice Sheet using BISICLES (with the same linear viscous relation of $m = 1$ as in this study), for which basal traction coefficients were inverted from ice surface velocity data (Cornford et al., 2015). These values are detailed in section 4.

### 3.4   Basal fluxes

The basal heat flux is set to a constant 50 mW $m^{-2}$ following the methodology of Gregoire et al. (2012), which ensures consistency with the initial ice temperature fields (also taken from the same study). They chose the value based on a geothermal map of North America (Blackwell and Steele, 1992), which is indicative of a fairly homogeneous modern geothermal heat flux under the modelled area. The value is of similar magnitude to modern-day geothermal heat flux under the Antarctic Ice Sheet, which has recently been suggested to vary between 42–180 $\mathrm{mWm}^{-2}$, with a mean of 68 mW $\mathrm{m}^{-2}$ (Martos et al., 2017). If all of the available heat from the basal heat flux of 50 mW $\mathrm{m}^{-2}$ would result in ice melt, the ice at the bottom would melt at a rate of ~5 mm $\mathrm{a}^{-1}$. Since this value is 2–3 orders of magnitude smaller than the surface melt rates, we assume for the purposes of this study that it is negligible and the basal melting rate is set to 0 m $\mathrm{a}^{-1}$ in all simulations. While basal and frictional heat fluxes do not directly result in ice melt at the base in these simulations, they do affect the ice flow by changing the effective
viscosity at the base of the ice sheet, as discussed in Section 2.1.

We chose to impose sub-shelf melt rates as a constant that is uniform in space and time. A value of 15 m $\mathrm{a}^{-1}$ was chosen for the sub-shelf melt rate for the 'standard' simulation. The chosen value is comparable to areas of high sub-shelf melt around modern-day Antarctic Ice Sheet, where the rates have been estimated to be up to ~43 m $\mathrm{a}^{-1}$ (Rignot et al., 2013). Using a single uniform and constant value for the sub-shelf melt over the whole model period is a simplification of the process, whereas in
reality the temperature of water in contact with the ice shelf likely underwent changes over the 10–8 ka period. Sub-shelf melt rates have been shown to be strongly positively correlated with increasing ocean temperatures, with a relationship of ~10 m $\mathrm{K}^{-1}$ estimated from radar interferometry -based observations of basal melt rates seaward of Antarctic grounding lines and nearby in-situ ocean temperature measurements (Rignot and Jacobs, 2002). It is, however, unclear if the relationship between the sea surface temperature in the Labrador Sea and sub-shelf melt rate could capture the circulation at the main marine margin of the
LIS at the time, as the ice sheet melt likely had a strong influence on the water density, temperature and circulation within the Hudson Strait. There is still limited scientific understanding of how sub-shelf melt rates vary as a function of location, shapes of cavities under the shelves, ocean circulation and ocean temperatures. To account for these large uncertainties, we tested a wide range of values (2–45 m $\mathrm{a}^{-1}$) for the sub-shelf melt rate as part of the sensitivity study.

### 3.5   Surface mass balance

The model defines SMB using the PDD scheme described in section 2.4. Values of 4.5 and 12 mm $\mathrm{d}^{-1}$ $°\mathrm{C}^{-1}$ were chosen respectively for the PDD factors of snow and ice while setting up the 'standard' simulation. These values are 50% higher than the typically used values of 3 and 8 mm $\mathrm{d}^{-1}$ $°\mathrm{C}^{-1}$, and were chosen by manual tuning for the 'standard' as a stronger ablation is necessary for the LIS to deglaciate in accordance with the reconstructed ice extent (Dyke, 2004). The need for higher PDD

factors was apparent as the ice sheet underwent substantial growth in the early simulations that employed the typical PDD factors, and the total volume of the ice sheet was still higher than the initialised ice volume after 1500 years of simulation. This may in part be related to BISICLES correcting the fact that the initial ice sheet (which follows ICE-6G_c) is too small in volume for its respective aerial extent (see section 3.2). However the main driver for the growth is the PDD method, which

has previously been highlighted as challenging for modelling the SMB of palaeo ice sheets (Bauer and Ganopolski, 2017; Charbit et al., 2013; Van de Berg et al., 2011), with one major problem being that the method does not explicitly account for the absorption of shortwave radiation (e.g. Van de Berg et al., 2011). The fixed PDD factors also do not take into account the temporal and spatial variability in insolation, cloud optical properties, snow properties, snow density, water content in snow and changes in albedo, among other assumptions (Charbit et al., 2013). Following this, recent research has suggested

that using the standard values is unlikely to predict the melt rates correctly under climatic conditions that differ from those in present-day Greenland (Bauer and Ganopolski, 2017; Charbit et al., 2013; Van de Berg et al., 2011). Bauer and Ganopolski (2017) compared the use of a physically-based surface energy balance method and the PDD method for simulating the North American Ice Sheet over the last glacial cycle, and found that PDD factors of 9 and 16 mm $d^{-1}$ $°C^{-1}$ respectively for snow and ice produce the best results for 15 ka conditions (the study did not provide an estimate for a time period closer to the timing of

the Hudson Bay saddle collapse). Moreover, exactly what the 'best' values for the PDD factors are is dependent on the model setup, which highlights the importance of assessing this parametric uncertainty through a sensitivity study.

The temperature lapse rate $\gamma$ is set to 5 °C $km^{-1}$ following the experimental setup of Gregoire et al. (2012). The value is based on the numerical modelling work of Abe-Ouchi et al. (2007) who found it more appropriate for the North American Ice Sheet than the typically used range of 6–8 °C $km^{-1}$.

## 4  Ice sheet sensitivity to model parameters

To gain a better understanding of which parameters control the rates of LIS deglaciation and the magnitude, duration and timing of the Hudson Bay saddle collapse, a series of sensitivity simulations were performed. The simulation labelled 'standard' is used as the starting point for varying individual parameters, and the parameters are varied systematically, one at a time. This approach allows for examining the effects of adjusting individual parameters while keeping the rest constant. This is particularly

important because BISICLES has not previously been used in a palaeo context for the Laurentide Ice Sheet, for which these values cannot be obtained using direct observations or inverse methods. Two simulations were run to assess the sensitivity to each of the 5 model parameters, resulting in a total of 11 simulations with the model parameters and their respective ranges shown in Table 1. To evaluate the impact of the transient climate and the spin-up adjustment period, a 'control' simulation with a constant 10 ka climate and 'standard' set of parameter values was also included.

### 4.1  LIS evolution during the initial adjustment period

This section describes the changes in the 'standard' simulation during the initial adjustment period over 10–9 ka. This spinup period results in substantial changes in ice sheet topography and adjustments to ice velocity. This is largely because the cho-

**Table 1.** Model parameters varied in the study and their ranges, with 'standard' values (as discussed in section 3) shown in brackets. The reference column indicates sections of the text discussing the specific model parameter.

| Parameter | Symbol | Value ('*standard*') | Unit | Reference |
|---|---|---|---|---|
| Levels of refinement | $\Omega^{0-2}$ | 10–2.5 (5) | km (grid size) | 3.1 |
| Basal traction | C | 1-6× 'standard' | Pa m$^{-1}$ a | 3.3 |
| PDD factor for snow | $\alpha_s$ | 3–6, (4.5) | mm (d K)$^{-1}$ | 3.5 |
| PDD factor for ice | $\alpha_i$ | 8–16 (12) | mm (d K)$^{-1}$ | 3.5 |
| Sub-shelf melt rate | $M_{ss}$ | 2–45 (15) | m a$^{-1}$ | 3.4 |
| Precipitation | P | 0.5–1× 'standard' | kg m$^{-2}$ d$^{-1}$ | 2.4 |

sen initial ice thickness field from ICE-6G_c is not physically consistent with the dynamical BISICLES ice sheet model, as discussed in section 3.2. The ICE-6G_c includes "unphysical geometric details" (Stuhne and Peltier, 2017), because the reconstruction is built from glacial isostatic adjustment (GIA) measurements, LIS extent data (Dyke, 2004), a model of viscoelasticity (VM5a) and relative sea level records for estimating the volumetric distribution of the ice mass (Peltier et al., 2015). This ad-

justment of the simulated ice sheet is therefore both necessary and expected. The heterogeneous pattern of depth-integrated ice velocities after 50 years of simulation (fig. 3b) illustrates this ongoing adjustment.

Consistent with the reconstruction it was initialised from, the ice sheet starts off with a small volume compared to what would be expected based on the mapped extent (see Section 3.2). The majority of these extensive parts of the ice sheet are initialised at 0 degrees Celsius, which is a very high temperature for ice (fig. 2). This promotes the ice flow through eq. 2,

but does not affect the surface mass balance since the accumulation and ablation are defined separately using the PDD model (Section 2.4; Rutt et al., 2009). The thin extensive parts of the ice sheet (Figures 2; difference between panels b and c in fig. 3) retreat within the first hundred years of simulation due to the ice either reaching flotation and calving or melting, or the low surface elevation leading to high melt rates. This may be an indication that the reconstructed ice in these regions is too thin. As a result, the extent decreases by ~37 % from $6.08 \cdot 10^6$ km$^2$ to $3.86 \cdot 10^6$ km$^2$ during the spin-up phase (10-9 ka). It is

worth noting that the climate forcing used in these simulations shares the ice mask with the ICE-6G_c reconstruction (Section 2.4; Peltier et al., 2015), and thus the simulated extent is mostly in agreement with the reconstruction, as will be discussed in Section 4.2. The evolution of ice volume in 'standard' undergoes less relative change than the extent during the spinup, with the ice loss through ablation and dynamic ice export mostly being balanced by accumulation. In other words, while the thin marginal parts of the ice sheet are retreating, some parts in the interior of the ice sheet thicken during the spin-up. This is

particularly noticeable in the saddle between the Keewatin and Labrador dome over Hudson Bay. By 9 ka the simulated LIS volume above flotation decreases by approximately 7 %.

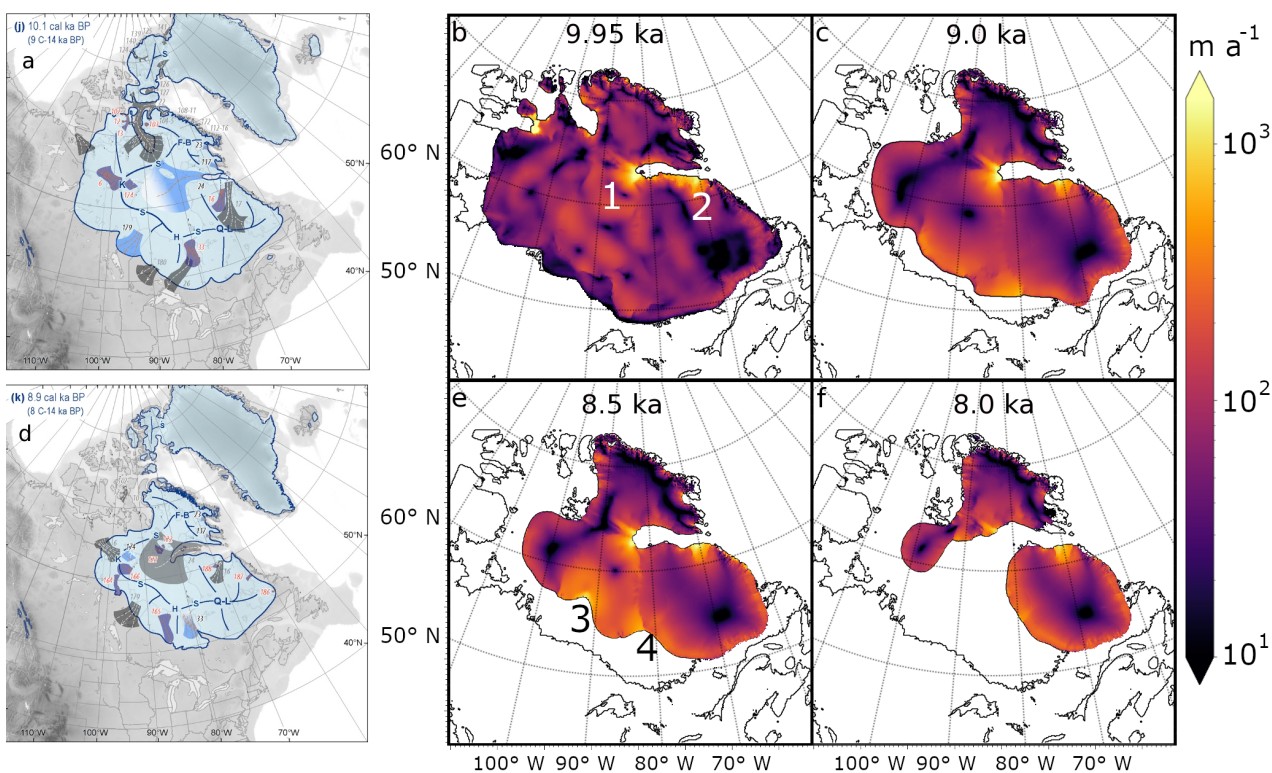

**Figure 3.** Comparison of modelled and observed ice stream activity. Reconstruction of LIS ice stream activity at 10.1 ka **a** and 8.9 ka **d** (adapted from Figure 5 in Margold et al., 2018). The ice streams that were active at the respective times are shown in light blue, ice streams that 'switched off' in the preceding 1000 years are shown in grey and those that 'switched on' shown in dark blue. Panels **b**, **c**, **e** and **f** show the magnitude of depth-integrated velocities ($ma^{-1}$) in the 'standard' simulation at 9.95 ka, 9.00 ka, 8.5 ka and at the end of the simulation (8.00 ka) respectively. In the panels showing the simulated ice velocities, modern coastlines are indicated by the black contour. The numbers 1 & 2 in panel **b** and numbers 3 & 4 in panel **e** show the approximate locations of four major reconstructed ice streams, and correspond to locations numbered 24, 16, 179 and 33 in fig. 5 in Margold et al. (2018) respectively.

## 4.2 Laurentide Ice Sheet evolution in 'standard' and 'control' simulations

This subsection describes how the ice sheet evolves in the 'standard' simulation, focusing on the 9–8 ka period. The simulated ice flow by 9.0 ka, 8.5 ka and 8.0 ka follows a pattern with smaller velocities at the ice domes, and faster-flowing ice over the central Hudson Bay, at the marine margins and towards the edges of the ice domes (fig. 3c, e & f). A reconstruction of LIS ice stream locations and their estimated timings (fig. 3a and d; Margold et al., 2018) allows for comparison of the simulated ice flow with data. The locations of reconstructed ice streams at the mouth of Hudson Strait and the Ungawa Bay (approximate locations numbered with 1 and 2 respectively in fig. 3b) stand out in the simulations with ice velocities consistently higher than 100 $ma^-1$ until the region in question deglaciates. The southwestern part of the ice sheet becomes dynamically more

active as more of the ice sheet becomes afloat and the ice sheet recedes over the softer sediments in Hudson Bay. This agrees well with contemporary ice shelves typically flowing faster than grounded ice (e.g. Rignot et al., 2011). The simulated pattern of ice flow is less organised in clear ice streams over the southwestern Hudson Bay than in Margold et al. (2018), although the highest ice velocities do coincide with the reconstructed Hayes Lobe and James Bay ice streams (approximate locations

numbered with 3 and 4 in fig. 3e). The simulated ice dynamics are potentially overestimated at the southwestern lake margin, but geomorphological evidence of ice streams in the region could also be less clear than over bedrock due to the ice flow being partially detached from the base and the softer sediments being more prone to reworking by the lacustrine and marine influence after the ice sheet receded further towards central Hudson Bay. The large coverage of fast-flowing parts of the ice sheet over the simulation (fig. 3) highlights the importance of ice export over the deglaciation.

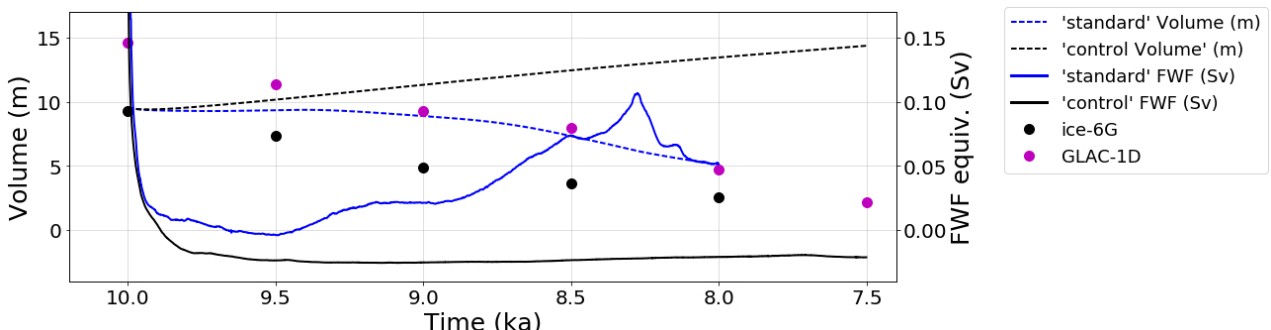

**Figure 4.** Evolution of ice sheet volume in metres of sea level rise equivalent (dashed line) and freshwater flux (FWF) equivalent in Sverdrups (solid lines). The volumetric SLR equivalent of the ice sheet is calculated from the volume above flotation, and the FWF from the total volumetric change between model years. The volume and FWF in the 'standard' simulation are shown in blue, and for the 'control' simulation in black. The black and magenta markers show the volume in metres of equivalent SLR in the ICE-6G_c (VM5a; Peltier et al., 2015), and the GLAC-1D (Tarasov et al., 2012) reconstructions.

The volumetric loss and the elevated freshwater flux (FWF) over the first ~100 years in both 'control' and 'standard' simulations (fig. 4) is a result of the modelled ice sheet undergoing dynamic reorganisation after initialisation and initially high ice melt rate of the extensive parts of the ice sheet, mainly on the southern and northwestern parts of the simulated LIS (fig. 5). The subsequent increase in 'control' ice volume shows that the ice sheet is not in equilibrium with the 10 ka climatic forcing. The evolution of the volume above flotation in the 'standard' simulation differs from the negative trend in the volumetric evolution

of the LIS in the two reconstructions shown in fig. 4, where the 'standard' simulation is compared to the ICE6G_c and GLAC-1D reconstructions. The volumetric change over 10–8 ka in the ICE-6G_c and GLAC-1D reconstructions are on average 0.34 m per 100 years and 0.50 m per 100 years respectively, and indicative of a continual decrease in volume over the time period.

The total volumetric change in the 'standard' simulation converted into FWF (solid blue line in fig. 4), shows a period of accelerated melt from 8.679 ka onward, with the melting defined as accelerated in the simulation over periods when the FWF

value is higher than 0.05 Sv (the value used for the background meltwater flux in Matero et al., 2017). The FWF reaches its peak value of 0.106 Sv during a 200 -year period of acceleration from 8.363 ka onward, and corresponds to the separation of

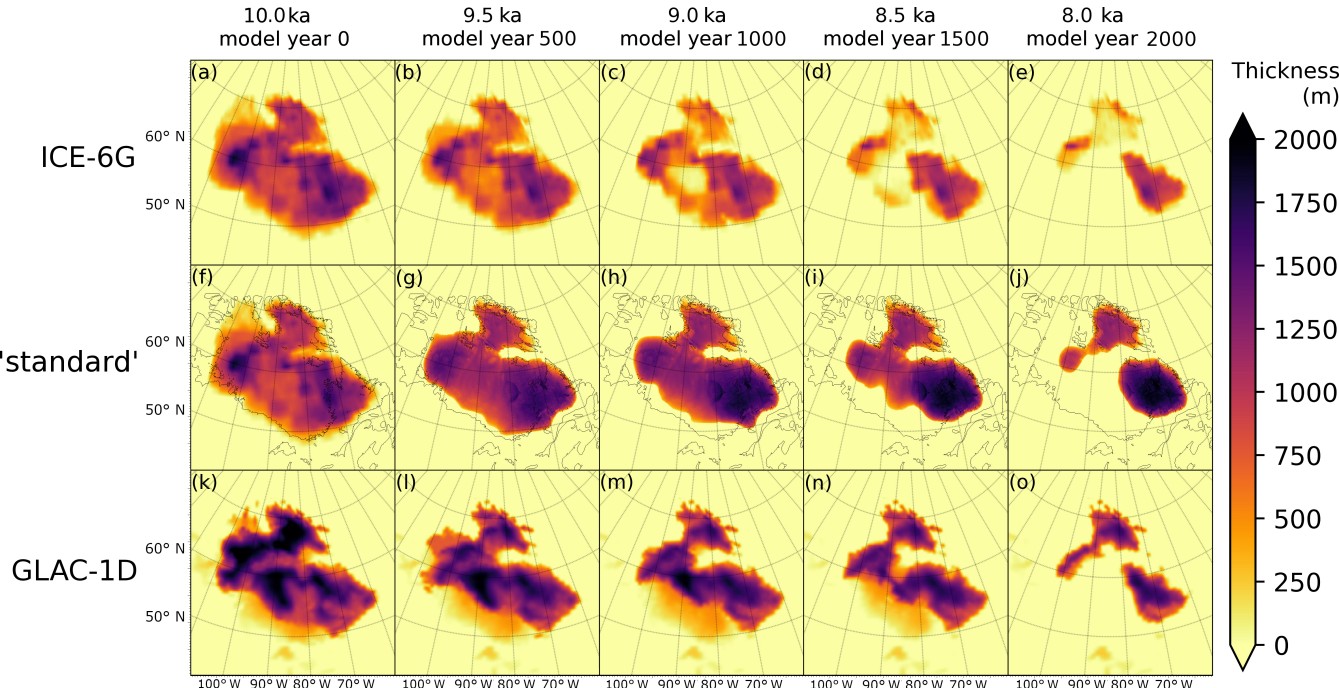

**Figure 5.** Ice sheet thickness evolution in the ICE-6G_c (VM5a; Peltier et al., 2015) reconstruction in panels **(a)**–**(e)**, the 'standard' simulation in panels **(f)**–**(j)** and the GLAC-1D reconstruction (Tarasov et al., 2012) in panels **(k)**–**(o)**. The ice thickness in each series is plotted in 500 year intervals. The coastlines plotted in panels **f**–**j** are based on the "ETOPO 10 ka" topography described in section 3.1.

the Labrador and Keewatin ice domes. The simulated separation of the ice domes occurs at a similar time to the scenarios in Matero et al. (2017), in which the peak FWF was released at approximately 8.25 ka. This is close to the timing that freshening signals from North Atlantic sediment cores suggest that the largest release of meltwater from the LIS would have taken place (~8.49 ka or ~8.29 ka, Ellison et al., 2006; ~8.38 ka, Kleiven et al., 2008; ~8.5 ka, Lochet et al., 2019). The timing and duration of the modelled saddle collapse in 'standard' are thus similar to some of our scenarios in Matero et al. (2017). However, the total SLR equivalent of the released FWF in 'standard' over the 200 -year period is 1.57 m, which is approximately 46% of the 3.39 m attributed to the saddle collapse period in the 4.24m_200yr scenario in Matero et al. (2017). Nonetheless, the simulated volumetric ice loss is close to the rate of volumetric change in the GLAC-1D ice sheet reconstruction over a wider 500-year window; 3.3 m of SLR equivalent from 8.5 to 8.0 ka (fig. 4).

The major changes over 10–9 ka in 'standard' are a ~37 % decrease in the ice extent from $6.08 \cdot 10^6$ km$^2$ to $3.86 \cdot 10^6$ km$^2$, and a reorganisation of the ice mass resulting in thickening of the ice sheet at the ice saddle and Labrador ice dome, and a thinning of the Keewatin ice dome (fig. 5f–j). It is also interesting to note that by 9 ka, the modelled ice sheet in 'standard' adjusts itself to resemble the GLAC-1D reconstruction more than the ICE-6G_c reconstruction in terms of volumetric evolution (fig. 4), shape and extent (fig. 5), despite the simulation initially being set up using the ICE-6G_c configuration. This is likely due to

the climatic influence on ice sheet evolution and the dynamical aspects of the BISICLES and GLAC-1D models, suggesting that these are important components for accurate representation of palaeo ice sheets.

One feature that particularly stands out as being different between the simulated ice sheet (fig. 5h) and the ICE-6G_c reconstruction (fig. 5c) by 9 ka is the thickness of ice over central Hudson Bay, which is the ice saddle connecting the Labrador and Keewatin ice domes. The GIA-based ICE-6G_c reconstruction indicates that the ice sheet would have deglaciated 'inside out', with the central part being free of ice before the surrounding regions. This pattern of deglaciation is not reproduced in any of the BISICLES simulations or the GLAC-1D reconstruction (fig. 5m). The difference in the volumetric change between the simulated ice sheet and the ICE-6G_c reconstruction (fig. 4) is largely a result of differences in modelled and reconstructed ice thickness over the Labrador and Foxe ice domes. These differences between the ICE-6G_c reconstruction and the model are clearest at 8 ka in (figures 5e and 5j respectively), with the modelled ice thickness having a maximum of 2188 metres at the Labrador dome compared to just over 1200 metres for ICE-6G_c.

### 4.3 Impact of varying the mesh refinement level

This subsection describes the results of increasing and decreasing the level of refinement of the adaptive mesh. Simulation 'AMR_0' had 0 levels of refinement ($10 \times 10$ km resolution, $\Omega^0$), 'standard' had 1 level of refinement (up to $5 \times 5$ km resolution, $\Omega^1$) and 'AMR_2' had 2 levels of refinement (up to $2.5 \times 2.5$ km resolution, $\Omega^2$). Running the current setup with 3 levels of refinement proved infeasible computationally, as run speeds decreased dramatically from the average of 396 model years per day (2 levels of refinement) to less than 10 model years per day (3 levels of refinement), and thus this part of the sensitivity study was limited to 2 levels of refinement.

Increasing the level of refinement of the AMR from $\Omega^0$ to $\Omega^2$ does not have nearly as big an impact on the long-term rates of change in volume (fig. 6a), as has been reported in a study of the West Antarctic Ice Sheet using BISICLES (Cornford et al., 2016), for example. Cornford et al. (2016) used a base resolution of 8 km, and increasing the refinement to 4 km or 2 km grid size resulted in distinct rates of change (see fig. 2 in Cornford et al., 2016), highlighting the necessity for using high resolution for simulating marine ice sheets. Based on these initial simulations, it seems that such high resolution is not as critical between these levels of refinement for the LIS deglaciation, possibly because it has a smaller marine or lake-terminating margin than West Antarctica (fig. 6a). These results do not, however, preclude the potential for differing patterns of deglaciation with further refinement (and alternate boundary conditions). Increasing the level of refinement from $\Omega^1$ to $\Omega^2$ results in the peak FWF occurring 19 years earlier, and decreasing the level of refinement from $\Omega^1$ to $\Omega^0$ results in the peak FWF occurring 4 years later. The durations and peak values in the discharge are similar between the three simulations, at 0.107, 0.107 and 0.106 Sv for 'AMR_0', 'standard' and 'AMR_2' respectively. The timing of the peak FWF in these simulations differs mainly between 8.45–8.11 ka, which coincides with the deglaciation of the part of the ice sheet connecting the Keewatin and Labrador domes over Hudson Bay (the saddle collapse).

Fig. 7 shows the impact of increasing the level of refinement of the adaptive mesh during the simulated saddle collapse at 8.3 ka. The highest level of refinement (2.5 km grid resolution in 'AMR_2') is automatically applied for the most dynamic regions, which in figure 7c is apparent over the centre of the ice saddle and the ice shelves. The overall ice flow in the ice saddle

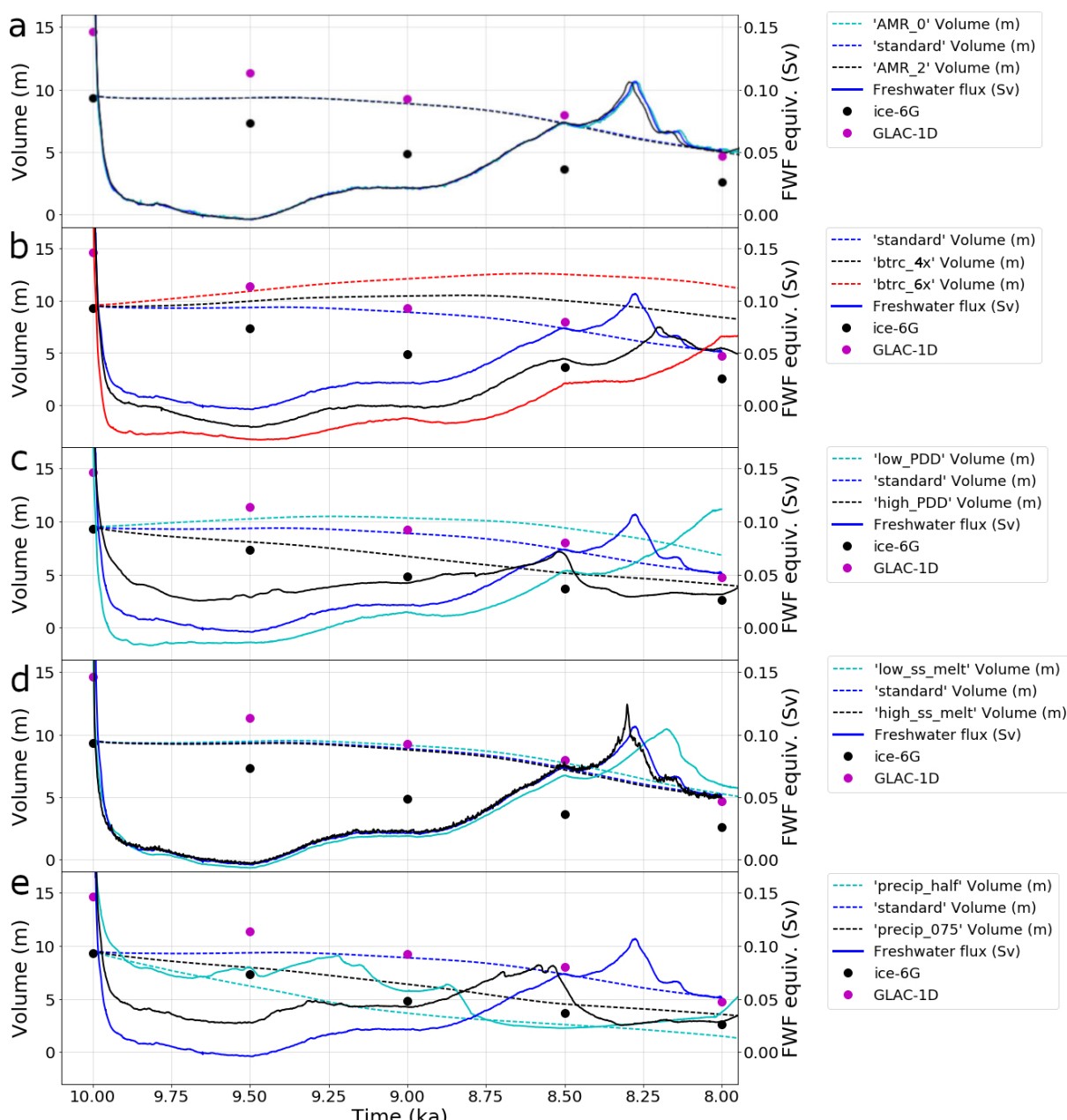

**Figure 6.** Effects of varying (**a**) adaptive mesh resolution, (**b**) basal traction coefficient C, (**c**) PDD (positive-degree-day coefficients), (**d**) sub-shelf melting rate and (**e**) precipitation on simulated ice sheet volume in meters of sea level rise equivalent, and freshwater flux equivalent in Sverdrups. The black and magenta markers show the volume in metres of equivalent sea level rise (SLR) in the ICE-6G_c (VM5a; Peltier et al., 2015), and the GLAC-1D (Tarasov et al., 2012) reconstructions.

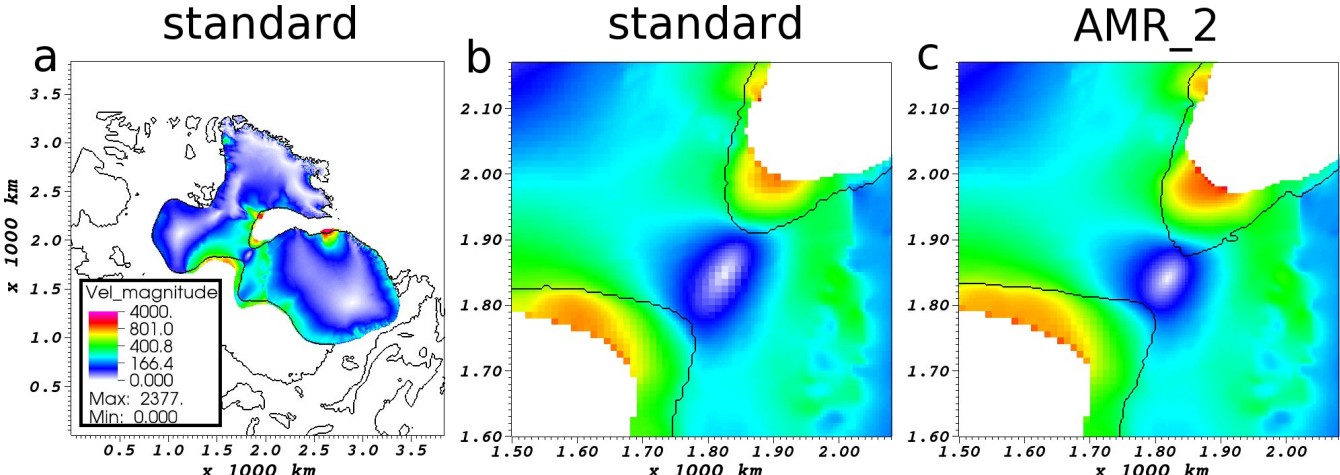

**Figure 7.** Magnitude of vertically integrated ice velocity (m a$^{-1}$) at model year 1700 with the grounding line and coastlines shown in black. The three panels follow the scale shown in panel **a**. **(a)** The velocity over the whole domain in the 'standard' simulation. **(b)** A zoomed-in view of the ice saddle in 'standard', and **(c)** in 'AMR_2' simulations.

is accelerated in the 'AMR_2' simulation compared to 'standard', with the fastest-flowing regions (shown in red in fig. 7) also being more extensive. The faster flow results in accelerated ice export towards the calving fronts and at 8.3 ka the grounding line on both sides of the ice saddle has retreated approximately 40 km further towards the centre in 'AMR_2' compared to 'standard', extending the ice shelves on both sides.

There is no clear difference in the magnitude of volumetric change between the simulations, and the difference in timing of the peak meltwater flux between the 'AMR_0', 'standard' and 'AMR_2' simulations is small (fig. 6a). This suggests that even the lowest resolution with no mesh refinement ($\Omega^0$) is sufficient for making a first-order evaluation of the associated meltwater pulse, due to majority of the deglaciation over the 10–8 ka period being driven by a surface mass balance feedback in the simulations. Further refinement, such as simulating a realistic meltwater flux with up to decadal temporal resolution

does rely on producing a realistic simulation of the movement of the grounding line and surface elevation that both depend on accurate representation of ice dynamics. The resolution of the mesh is thus an important parameter to investigate in terms of understanding the saddle collapse, as increasing the level of refinement could further alter the deglaciation pattern of the ice saddle and ice streams.

### 4.4   Impact of varying the basal traction coefficient

The basal traction coefficient $C$ in the 'standard' simulation is defined as shown in fig. 1b, and $C$-values are set to 50 Pa m$^{-1}$ a for the sediment-covered regions, and to 80 Pa m$^{-1}$ a for the regions with submerged bedrock at the mouth of Hudson Strait. The $C$-values for the bedrock regions are defined as 400 Pa m$^{-1}$ a at sea level ($z$=0 m), and increase with the elevation of the bed at a rate of 150 Pa m$^{-1}$ a per 500 m. In the 'btrc_4x' simulation the $C$-values and rate of change with elevation of

the bed are quadrupled from the 'standard' values (and multiplied by 6 in the 'btrc_6x' simulation). The resulting $C$-ranges in the three simulations are 0–939 Pa m$^{-1}$ a, 0–3756 Pa m$^{-1}$ a and 0–5634 Pa m$^{-1}$ a for 'standard', 'btrc_4x' and 'btrc_6x' respectively. The effect of halving the basal traction coefficient values from 'standard' was also tested. This effectively makes the ice sheet base very slippery and causes the model to crash after 15 years due to extreme ice velocity acceleration that

becomes unsolvable.

    Increasing the basal traction between the simulations results in a near-uniform deceleration of the ice flow shortly after initialisation, which in turn slows down the export of ice from the domes and the transport of ice towards the ice margins. This, in combination with the high accumulation rates in the simulations, initially results in glaciation instead of deglaciation for the 'btrc_4x' and 'btrc_6x' simulations (red and black lines in fig. 6b). The peak freshwater flux in 'btrc_4x' also occurs 50 years

later (and 250 years later in 'btrc_6x') with smaller magnitudes and longer durations for the elevated meltwater flux compared to the 'standard' simulation. Panels **a–c** in fig. 8 show the ice thickness at 8.25 ka in each of the simulations, demonstrating that increasing the basal traction coefficient value results in thicker and more extensive ice domes.

    The resulting ice velocities are, however, high in 'standard' compared to modern velocities for the Antarctic Ice Sheet (e.g. Rignot et al., 2011, 2008; Rignot and Kanagaratnam, 2006) and in Greenland (Rignot and Kanagaratnam, 2006). This is likely

a result of the basal traction coefficient values in 'standard' being smaller than those solved using inverse methods for use with BISICLES for the modern West Antarctic Ice Sheet (Cornford et al., 2015). These earlier studies and the fact that the ice velocities in 'standard' are approaching the threshold of unrealistic ice velocities, suggest that the basal traction coefficient values used in these simulations could have been set to be too low to compensate for the high rates of ice accumulation. Another reason for the high velocities could be high stress and strain rates that result from the initial shape and surface slope gradients

of the ice sheet being at least initially too steep, having been initialised from the GIA-based ICE-6G_c reconstruction.

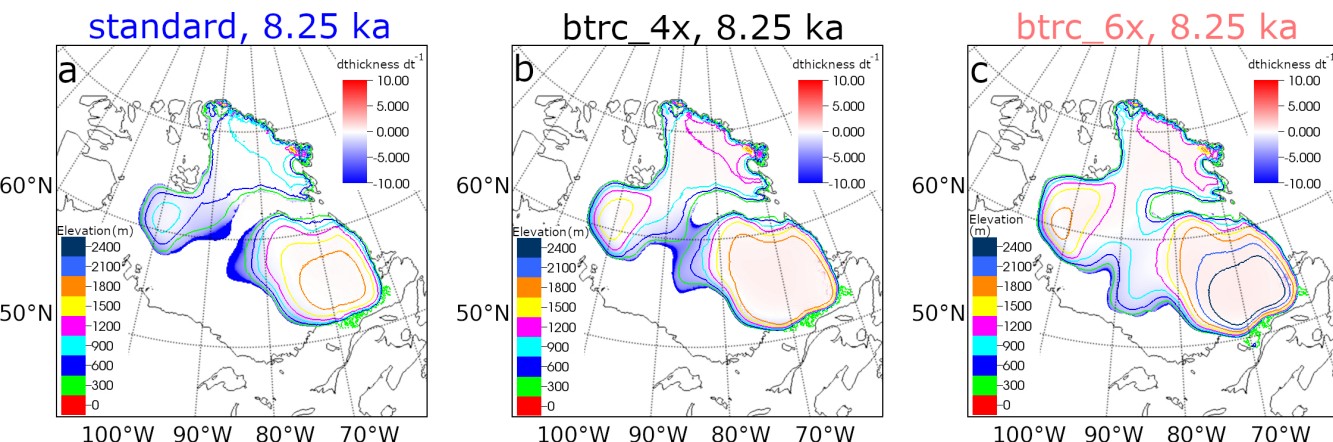

**Figure 8.** Modelled ice thickness and rate of change of thickness (m a$^{-1}$) at 8.25 ka in (a) the 'standard' simulation, (b) the 'btrc_4x simulation in which the standard basal traction coefficient field is quadruple, and (c) 'btrc_6x' in which the standard basal traction coefficient field is multiplied by 6.

In the model, there is still fast-flowing ice present at all three domes at 9 ka, with integrated velocities in the range of $10^2$ m a$^{-1}$. This magnitude is more characteristic for contemporary outlet glaciers in the Greenland Ice Sheet (Rignot and Kanagaratnam, 2006), but the LIS was experiencing rapid deglaciation at the time, and it is feasible that the ice flow rates towards the periphery were high. The presence of meltwater during the melting season has been shown to accelerate ice flow in the Greenland Ice Sheet (the "Zwally effect"; Zwally et al., 2002), and meltwater was likely extremely abundant during the surface-melt driven retreat of the LIS (Carlson et al., 2009). For comparison, the background FWF used in Matero et al. (2017) to represent the melting of the LIS outside the saddle collapse period was ~16 times the freshwater flux of 0.003 Sv from the contemporary Greenland Ice Sheet (Shepherd and Wingham, 2007), and this does not include the meltwater pulse from the Hudson Bay ice saddle collapse.

## 4.5 Impact of varying the PDD factors

The PDD factors in the 'standard' simulation are set to $\alpha_s = 0.0045$ mm (d K)$^{-1}$ for snow and to $\alpha_i = 0.012$ mm (d K)$^{-1}$ for ice. Simulation 'low_PDD' has lower PDD factors of $\alpha_s = 0.003$ mm (d K)$^{-1}$ and $\alpha_i = 0.008$ mm (d K)$^{-1}$, and for simulation 'high_PDD' these were set to $\alpha_s = 0.006$ mm (d K)$^{-1}$ and $\alpha_i = 0.016$ mm (d K)$^{-1}$.

Since the other climatic parameters are the same between the simulations, the initial impact of changing the PDD factors was expected to be fairly straightforward, with higher values resulting in more pronounced melting, which fig. 6c demonstrates. The surface melt rates in simulations with higher PDD factors include two important positive feedbacks for individual locations. Firstly, faster melting of the snow cover due to an increase in $\alpha_s$ has the compounded effect of accelerating the total surface melt once the snow cover melts completely due to $\alpha_i$ having a larger value than $\alpha_s$. Secondly, surface melt leads to lowering of the surface, which further accelerates the melting through local increase in the surface air temperature in accordance with the SAT lapse rate $\Gamma$.

The higher PDD factors in 'high_PDD' compared to the 'standard' result in the peak freshwater flux and the saddle collapse occurring 225 years earlier (fig. 6c), with a lower magnitude (0.07 Sv and 0.11 Sv respectively, fig. 6c). The separation of the Keewatin and Labrador ice domes in 'low_PDD' is delayed by over 275 years compared to the 'standard' simulation, and fig. 9a shows the ice sheet at the end of the simulation.

The different PDD factors between the simulations cause distinctly different patterns for the evolution of the ice volume above flotation in the simulations (fig. 6c). Over the first 1000 years of simulation, the volume of the ice sheet (calculated in meters of sea level rise equivalent) increases by ~8% in 'low_PDD', decreases by ~8% in 'standard' or decreases by 30% in the 'high_PDD' simulation, making the model setup highly sensitive to the value chosen for this parameter. 'high_PDD' is the first of the simulations presented here that is approaching a rate of volumetric change that is comparable to that of ICE-6G_c, in which the LIS volume decreases by an average of ~0.33 metres of SLR equivalent every 100 years for the period between 10 ka and 8 ka (0.28 m of SLR equivalent per 100 years in 'high_PDD'). In GLAC-1D, the average volumetric loss over the 2000 year period is approximately 0.50 metres of SLR equivalent per 100 years, which is 80% larger than the ice loss rate in 'high_PDD'. It is worth noting that the two are not directly comparable due to the different initial ice sheet geometries and ice volumes. The initial ice volume in 'high_PDD' is approximately two thirds of the volume in GLAC-1D at 10 ka, with

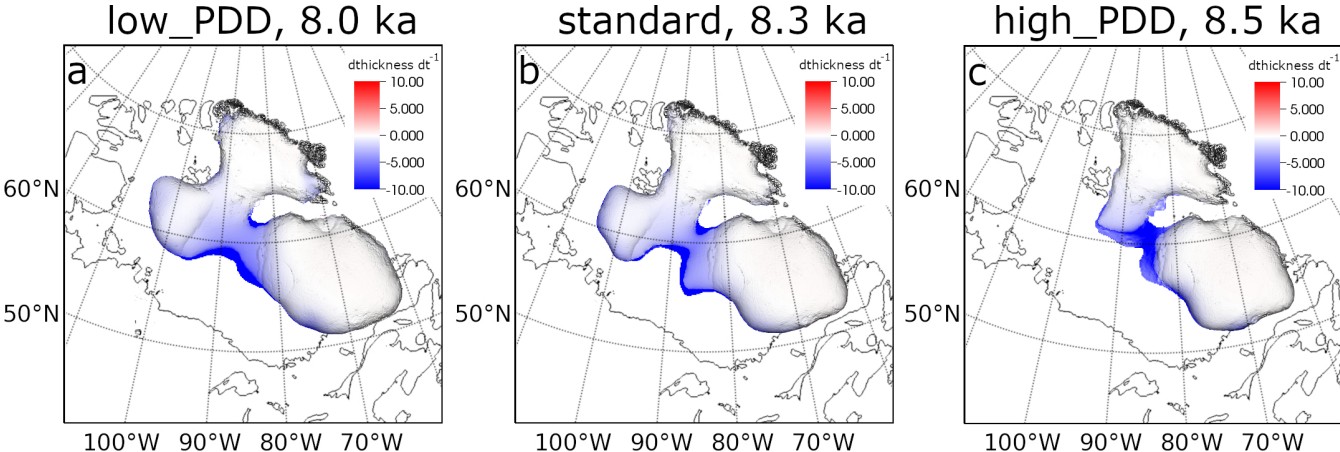

**Figure 9.** Laurentide Ice Sheet at the approximate time of the peak freshwater flux in the **(a)** 'low_PDD' simulation, **(b)** 'standard' simulation, **(c)** 'high_PDD' simulation. The separation of the Keewatin and Labrador domes in the 'standard' and 'high_PDD' simulations is at a more advanced stage at the time of peak freshwater flux compared to the 'low_PDD' simulation. The freshwater flux in 'low_PDD' would likely have increased as the separation of the ice domes would have continued if the simulation was run for longer than 2000 years.

the GLAC-1D ice sheet being thicker over a comparable extent (figures 5f & 5k). Both reconstructions indicate a more rapid fractional loss over the period than the BISICLES simulations presented here (fig. 6c), and in all three of our simulations, the Labrador dome is a stable feature and a constant store of freshwater by 8 ka (~3.71 *m*, 4.50 *m* and 4.58 *m* of SLR for 'high_PDD', 'standard' and 'low_PDD' respectively).

### 4.6 Impact of varying the sub-shelf melting rate

Three values for sub-ice shelf melting rate ($M_{ss}$) were tested in order to evaluate the sensitivity of the early-Holocene LIS deglaciation to this parameter: 5 m a$^{-1}$ ('low_ss_melt'), 15 m a$^{-1}$ 'standard' and 45 m a$^{-1}$ 'high_ss_melt'. Representing the sub-shelf melt with a single value over 10–8 ka is a simplification, as it is a process varying both in time and space even for individual ice shelves. An example of the possible spatial variability is a study by Rignot and Steffen (2008), who found the sub-shelf melt rate under Petermann Glacier in Northern Greenland to be highly channelised along the flowline, reaching between 0–25 m a$^{-1}$ over the 2002–2005 period. Seasonal variability, and whether the ice front at the marine terminus is an extensive ice shelf or a vertical calving front also has an impact on submarine melt rates. Indeed, individual tidewater glaciers have been estimated to undergo periods of extremely high summer melt at $3.9 \pm 0.8$ m day$^{-1}$ in Western Greenland (Rignot et al., 2010) and up to 12 m day$^{-1}$ at the Leconte Glacier in Alaska (Motyka et al., 2003). A source of uncertainty in these simulations is treating the lacustrine front on the southwestern side of the LIS (i.e. Lake Agassiz) with the same values for basal ice sheet melt as the marine margins. The Lake Agassiz sub-shelf melt is currently difficult to constrain due to both the volume and extent of the lake being uncertain over time (Leverington et al., 2002; Clarke et al., 2004) and no studies have been published on the potential heat budget of the lake and its interactions with the LIS.

The evolution of ice volume in the two simulations with larger sub-shelf melt values ('standard' and 'high_ss_melt') is similar until 8.35 ka, when larger regions of the ice saddle connecting the Labrador and Keewatin domes thin sufficiently to become afloat (fig. 6d). Following this, the deglaciation of the central Hudson Bay in 'high_ss_melt' accelerates in comparison to 'standard', resulting in a peak meltwater flux of 0.124 Sv that occurs 24 years earlier than the peak flux of 0.107 Sv in 'standard'. The difference in timing of the peak meltwater discharge between the 'low_ss_melt' and 'standard' simulations is larger (101 years), but the values of peak discharge between the two simulations are very similar. In addition to being delayed, the saddle collapse meltwater pulse in 'low_ss_melt' has a longer duration.

For the majority of the 2000 -year simulation the rate of volumetric change of the LIS is not sensitive to varying the sub-shelf melt, but the parameter becomes more important during the more dynamic part of the deglaciation once parts of the ice sheet over Hudson Bay thin sufficiently to begin to float. The rate of meltwater flux in 'low_ss_melt' starts to deviate from that of the two simulations with higher sub-shelf melt rates after ~8.72 ka, which is likely due to the ice shelves in 'low_ss_melt' exerting a stronger buttressing effect on the ice flow and export across the grounding lines at the marine margins. An interesting piece of future work could be to study the importance of sub-shelf melt rates together with increasing the model resolution to sub-kilometre grid cell size, and to examine the changes in the Hudson Strait ice stream and movement of the grounding line there. Another potential development could be to allow for temporal evolution of the sub-shelf melt. At the time of the LIS demise, the orbital configuration of the Earth was approaching the Holocene Climatic Optimum conditions post 10 ka (Kaufman et al., 2004). The associated increase in radiative forcing, together with changes in Atlantic Meridional Overturning Circulation and associated meridional heat transport (Ayache et al., 2018), could have had an impact on the heat absorption to the lake and sea water adjacent to the ice sheet over the modelled period. Their impact would, however, likely be minor due to the low sensitivity shown in response to the large range of sub-shelf melt parameters tested in this study.

## 4.7  Impact of varying the precipitation rates

The input climatologies to the ice sheet model contain significant uncertainty as there are no observations for precipitation and temperature over the ice sheet. For practicality, precipitation values for the 'standard' run were taken directly from the HadCM3 deglaciation snapshots (section 2.4), but this is the output from only one climate model. Indeed, the current generation of GCMs (e.g. Taylor et al., 2012) shows large regional variability in the climatic response to mid-Holocene settings (Harrison et al., 2015), and the GCMs participating in the second phase of the Palaeoclimate Modelling Intercomparison Project (PMIP2 Braconnot et al., 2007) indicated a wet bias for eastern North America (Braconnot et al., 2012). At the time of setting up this study, other climate model output was available and at similar spatial resolution, (e.g. Liu et al. (2009)), but the HadCM3 results are currently the only climate fields produced using the latest boundary conditions for the period, including the ICE-6G_c reconstruction that we initialised our ice sheet from. In fact, HadCM3 has been shown to perform well at simulating the period (see discussion within Morris et al., 2018, Supporting Information), yet even with the latest protocol, there is uncertainty in the model boundary conditions used to run the climate simulations (e.g. see Ivanovic et al., 2016). Ice sheet topography in the GCM simulations (ICE-6G_c) is likely to be a particular source of error, with precipitation being negatively correlated with elevation (Bonacina et al., 1945). For context, the LIS in ICE-6G_c and GLAC-1D reconstructions have distinctly different topographies

at 10 ka (fig. 5), with the three domes and the ice saddle being considerably lower in ICE-6G_c. However, the choice of using ICE-6G_c in the climate simulations is most consistent with our approach of initialising the ice sheet simulations from ICE-6G_c). A possible source of temporal uncertainty in the precipitation is that, and again for practical purposes, our forcing assumes smooth interpolation between the climate means spaced at 500-year intervals, which is unlikely to accurately represent the detailed evolution of North American climate even if it does capture the large-scale glacial-interglacial trends. Nevertheless, it is interesting to see how the ice sheet model responds to the gradual nature of the forcing and more detailed temporal precipitation fields from climate models using the latest and most physically robust boundary conditions are not yet available.

Any biases in the input precipitation can affect the surface mass balance, for example through anomalous accumulation of snow that transforms to ice, and the smaller PDD factor of snow compared to that of ice resulting in excessive snow cover slowing down surface melt. Thus, it is valuable to gain a more rigorous understanding of the impact of the input precipitation fields on our simulated ice sheet evolution. Fig. 10 shows the evolution of the ice sheet thickness in three simulations with different precipitation fields $P(t,x,y)$. For 'precip_0.75' the $P$ field in 'standard' is multiplied by 0.75 (25% reduction), and for 'precip_half' the $P$ field is halved while other model parameters are kept constant.

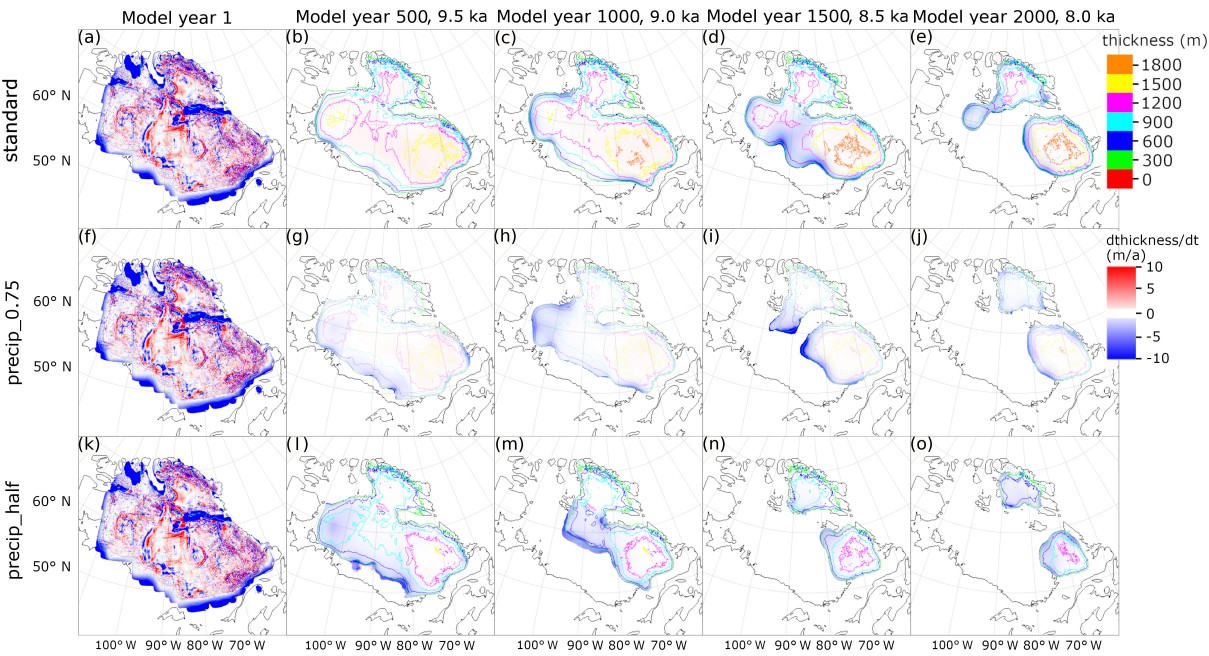

**Figure 10.** Laurentide Ice Sheet thickness evolution in the three simulations with varying precipitation fields in 500 -year intervals. **(a)**–**(e)** 'standard', **(f)**–**(j)** 'precip_0.75' (P field in 'standard' multiplied by 0.75), and **(k)**–**(o)** 'precip_half'(P field in 'standard' halved).

Scaling the input precipitation while keeping the temperature constant can be considered unphysical because the two fields are climatologically interdependent, with precipitation usually increasing with temperature (Trenberth and Shea, 2005; Harrison et al., 2015). However, it is useful to separate out the role of the two fields since they are not linearly related, and hence one of the objectives of this study is to assess the sensitivity of the model setup to individual parameters. Separating temperature

from precipitation in this idealised way allows for better understanding of the impact that the precipitation boundary condition has on simulated ice sheet evolution.

As described in the previous subsections, the accumulation of ice results in the growth of the Foxe and Labrador domes over the majority of the simulations. Decreasing the input precipitation produces a faster deglaciation of the southwestern parts of the ice sheet, namely the Keewatin dome and the modern southern Hudson Bay region.

The importance of the precipitation field is highlighted by the time series describing the volumetric changes of LIS over time (fig. 6e). The modelled separation of the Keewatin and Labrador domes and peak freshwater fluxes in simulations with smaller $P$ occur approximately 400 and 700 years earlier than in 'standard' for 'precip_0.75' and 'precip_half', respectively. The rate of ice loss also changes significantly as a result of decreasing precipitation (dashed lines in fig. 6e); already by 9 ka, approximately 65% (42%) of the total initial volume of ice has gone in 'precip_half' ('precip_0.75'), as opposed to approximately 15% of volumetric loss in 'standard'. Thus we can surmise that LIS is extremely sensitive to variations in the precipitation field and the resulting changes in surface mass balance.

## 5 Discussion

The simulated early-Holocene deglaciation of the LIS in 'standard' is in agreement with the sequence of parts of the ice sheet becoming ice-free and their timing is mainly within the reported error estimate of ~500–800 years with the empirical ice extent reconstruction of the North American Ice Sheet (Dyke, 2004, fig. 5). The rate of overall LIS ice loss differs from the GLAC-1D and ICE-6G_c reconstructions for the 10–9 ka period (fig. 4), but the simulated decrease in ice volume over the 9–8 ka period is close to the GLAC-1D reconstruction. The area covered by 8 ka, is within 20% of the reconstructions with extents of $2.36 \cdot 10^6$ $km^2$, $2.25 \cdot 10^6$ $km^2$ and $2.01 \cdot 10^6$ $km^2$ respectively for ICE-6G_c, GLAC-1D and the 'standard' simulation. The ice is thickest at 8 ka in 'standard', with the majority of the differences arising from the simulated Labrador dome ice volume (1.99 m, 2.63 m and 4.50 m of sea level rise equivalent respectively in ICE-6G_c, GLAC-1D and 'standard'). The Labrador dome ice volume at ~8.2 ka has recently been estimated at $3.6 \pm 0.4$ m of eustatic SLR (Ullman et al., 2016).

The 3.8 m ka$^{-1}$ volumetric change over the 10–8 ka period in the 'standard' simulation is smaller than the eustatic SLR of ~15 m ka$^{-1}$ for 11.4–8.2 ka based on sea level records (Lambeck et al., 2014). The majority of the SLR in the early Holocene has been attributed to the LIS, with an estimated Antarctic contribution of 0.25–0.3 m ka$^{-1}$ (Briggs and Tarasov, 2013). The simulated ice loss over the modelled period is also smaller than the estimated volumetric change in GLAC-1D & ICE-6G_c reconstructions (~5 m ka$^{-1}$ LIS contribution over 10–8 ka). The simulated ice volume in SLR equivalent at 8 ka is, however, nearer the estimated Labrador dome volume at ~8.2 ka (Ullman et al., 2016) than the ICE-6G_c estimate. This, together with the higher ice volume in GLAC-1D through 10–8 ka and the discussed low volume:extent ratio (section 3.2), suggests that the initial ice volume in the simulations could be underestimated.

A major feature that differs in the pattern of deglaciation between the ICE-6G_c and 'standard' is the thickening of the three ice domes and the ice saddle over 10–9 ka in the simulation, which results in a comparable ice volume by 9 ka (8.89 m SLR equivalent) with GLAC-1D (9.24 m SLR equivalent), both significantly higher than the ICE-6G_c estimate (4.84 m SLR

equivalent). Another pattern that is not present in 'standard' is the opening of the Tyrell Sea at ~9 ka in ICE-6G_c (figures 5c and 5h). Instead of the unrealistic opening of a hole in the middle of the ice sheet (which is how the Tyrell Sea opens in ICE-6G_c), BISICLES simulates accumulation and ice flow from the surrounding regions, resulting in thickening of the part of the ice sheet covering the Hudson Bay. At 8 ka, i.e. after the model has integrated for 2000 years (fig. 5j), the simulated

ice sheet is more similar to the GLAC-1D reconstruction (fig. 5o) than the ICE-6G_c reconstruction (fig. 5e). This similarity likely results from both the GLAC-1D reconstruction and these BISICLES simulations being based on dynamical ice sheet modelling and driven by a climate forcing, suggesting that having a dynamical component with a climate forcing is important for accurately reconstructing and modelling the LIS in order for the shape of the ice sheet to be physically consistent with ice dynamics.

Reaching a better understanding of the role of ice dynamics in the LIS deglaciation and the resulting freshwater flux was the motivation for setting up the set of experiments presented in this study. In 'standard', the distribution of the ice undergoes significant dynamic change over the spinup -period (10–9 ka; figures 3 & 5), while the volume of the ice sheet only decreases by 7%. This reorganisation is most prominent in the extensive thin areas of the LIS as well as in the interior of the ice sheet, which is being reshaped into three main domes around the Hudson Bay with the ice saddle in between. The surface elevation

in the interior of the ice sheet also undergoes change, transforming from a patchy structure (fig. 5f) into a smoother shape that is more consistent with ice sheet physics (fig. 5h). The dynamic ice loss balances the ice growth through snow accumulation over the course of the simulation, as demonstrated by the higher basal traction coefficient in 'btrc_6x' compared to 'standard' resulting in the ice volume growing until ~8.7 ka (figures 8a & c and 6b). The main ice export occurs at the calving fronts, at the lacustrine edge of the ice saddle and the defined ice streams at Ungawa Bay and at the mouth of Hudson Strait. It has

been proposed that the deglaciation of LIS was mainly driven by negative surface mass balance (e.g. Carlson et al., 2009), but it is important to note that the dynamics have an important role too, reshaping the domes as the areas with lower elevation ablate more rapidly due to the effect of the lapse rate. The importance of the ice export and having a good representation of ice streams through high-enough resolution and the inclusion of membrane stresses (as in BISICLES) are likely important in adequately representing the deglaciation and eventual demise of LIS (Gandy et al., 2019).

The base resolution of the model setup with 10 km grid size is already unprecedented in simulating the LIS deglaciation, and the application of the adaptive mesh allows for even finer spatial resolution at regions of interest in millennial-scale simulations, such as the moving grounding line and ice streams. An increase from one to two levels of refinement, corresponding to a smallest grid size of 5 km and 2.5 km respectively, results in ~40 km further retreat of the grounding line and more extensive ice shelves on both sides of the ice saddle by 8.3 ka (fig. 7). However, the impact of these dynamic changes on the magnitude

of the peak meltwater flux is small (fig. 6a), suggesting that already the lowest resolution is sufficient for first-order evaluation of the meltwater pulse associated with the ice saddle collapse. It is worth noting that we did not reach the aforementioned kilometre-scale resolution suggested as potentially required for adequately resolving marine interfaces of continental ice sheets (Favier et al., 2014; Cornford et al., 2016) due to the high computational cost of the simulations, particularly during the early stages of the adjustment period (first 1000 years of the simulation). Further refinement of the mesh could result in the ice saddle

reaching flotation earlier, with knock-on effects on the dynamics of the collapse.

**Table 2.** Peak FWF duration, amplitude and timing (ka) in each simulation. The peak is defined as ongoing when the amplitude is greater than the background flux of 0.05 Sv (as defined in section 4.2). The 'n/a' for simulations 'btrc_6x' & 'low_PDD' indicates that the peak of the saddle collapse does not occur prior to 8 ka. The reference column indicates sections of the text discussing the specific simulations.

| Parameter | Duration (a) | Amplitude (Sv) | Timing (ka) | Reference |
|-----------|--------------|----------------|-------------|-----------|
| 'standard' | 690 | 0.11 | 8.276 | 4.1 |
| 'AMR_0' | 726 | 0.11 | 8.271 | 4.2 |
| 'AMR_2' | 690 | 0.11 | 8.296 | 4.2 |
| 'btrc_4x' | 334 | 0.07 | 8.202 | 4.3 |
| 'btrc_6x' | n/a | n/a | n/a | 4.3 |
| 'low_PDD' | n/a | n/a | n/a | 4.4 |
| 'high_PDD' | 301 | 0.07 | 8.520 | 4.4 |
| 'low_ss_melt' | >705 | 0.11 | 8.176 | 4.5 |
| 'high_ss_melt' | 693 | 0.13 | 8.303 | 4.5 |
| 'precip_075' | 408 | 0.08 | 8.582 | 4.6 |
| 'precip_half' | 1198 | 0.09 | 9.222 | 4.6 |

The opening of the Hudson Bay by an ice saddle collapse is a feature of particular interest due to its potential role as a major forcing for the 8.2 ka event (Matero et al., 2017). The modelled opening of Hudson Bay in 'standard' occurs at 8.26 ka, which is close to the timing in GLAC-1D (8.2–8.1 ka) and coeval with ICE-6G_c (between 8.5–8.0 ka; fig. 5d). These dates are based on the opening of a completely ice-free corridor between the two domes. Exchange of water masses between the Tyrell Sea and Lake Agassiz likely commenced earlier, as soon as the ice saddle thinned sufficiently to reach flotation (~500 m, assuming the modelled Hudson Bay basal topography and that water level was at sea level on both sides of the ice saddle).

Out of the studied parameters, the most significant differences in terms of rate of volumetric loss arose from varying the basal traction coefficients, PDD factors and the amount of precipitation. The model setup is sensitive to perturbations related to the surface mass balance (Table 2), which highlights the importance of carefully defining the atmospheric boundary conditions. Higher basal traction coefficients generally result in slower deglaciation (section 4.4) due to less transport of ice towards lower altitudes where melting is more pronounced because of the temperature-elevation feedback, as well as less dynamic ice loss at the marine and lacustrine margins. Higher PDD factors (section 4.5) or alternatively less accumulation through lower precipitation (section 4.7) result in a more negative surface mass balance.

Note, that the importance of each parameter cannot be directly compared in a quantifiable way, as the ranges used for the sensitivity analysis represent something different for each parameter, and were choices based on previous studies. For example, changing the precipitation affects the whole domain, whereas varying the sub-shelf melt rate only affects the floating part of

the ice sheet. The sensitivity to different parameters is therefore unlikely to vary in a similar way as a response to halving or doubling a specific model parameter, but their relative importance and interactions between the parameters can be examined.

As a result of the SMB having such an important role in the simulations (see sections 4.5 and 4.7), under- or overestimating the precipitation or values for PDD-factors can have a big impact on the modelled behaviour of the ice sheet. Accurately representing the climate in general circulation models for time periods different from the present is challenging, and different GCMs project regionally heterogeneous patterns of precipitation and temperature for both the future (Knutti et al., 2010) and different time periods during the last deglaciation (e.g. Braconnot et al., 2012). For the mid-Holocene, the ensemble averages of GCMs participating in the second phase of the Palaeoclimate Modelling Intercomparison Project (PMIP2; Braconnot et al., 2012) indicated a wet bias for eastern North America compared to climate reconstructions (fig. 1 in Braconnot et al., 2012). While the fact that the models indicate wetter than reconstructed conditions east of Hudson Bay for the mid-Holocene does not imply that the same is true for the early Holocene, they do suggest that the model representation of precipitation from a single GCM includes significant uncertainty in the region. If the modelled precipitation rates are too high, this could have a knock-on effect resulting in other model parameters such as basal traction and the PDD factors having been tuned to compensate for an unrealistically high accumulation rate.

The large proglacial Lake Agassiz and its interactions with the ice sheet and the climate is another source of uncertainty for modelling the LIS deglaciation. The area and surface temperatures of the lake are poorly constrained, and refining its representation in the climate forcing would affect the availability of moisture for precipitation. Additionally, the lake level is set to sea level in our model setup, whereas the lake could have been up to 770 $m$ above sea level prior to its final drawdown (Teller et al., 2002). The elevated water body could, therefore, have accelerated the ice melt at the southwest margin of the ice sheet due to increased flotation subjecting a larger area of the ice sheet to sub-shelf melting. However, the model does not, distinguish between freshwater and marine margins, and freshwater calving is typically about an order of magnitude lower than for marine margins in comparable settings (Benn et al., 2007b, and references therein). Finally, Lake Agassiz potentially acted as a source of heat at the ice-water interface, as the bed of Lake Agassiz was sloped towards the ice sheet and the density maximum of freshwater is above the freezing point (as opposed to seawater). Significant absorption of shortwave radiation to the lake could thus have resulted in transport of heat towards the base of the ice sheet through a flow of the warmest and densest water masses, and could have acted as an additional driver of retreat of the lacustrine ice front.

## 6  Conclusions

This is the first application of the BISICLES ice sheet model (Cornford et al., 2013) in the new palaeo setting of North American Ice Sheet deglaciation, and is an effort to combine data from multiple sources, recognising that the input datasets have their inherent uncertainties, as does the geological evidence used for evaluating the performance of the model setup. Nonetheless, as a result of this sensitivity study, a model setup with ranges for key model parameters for early-Holocene LIS deglaciation experiments is established. The importance of accurately representing the LIS ice dynamics during the 10–8 ka period is highlighted by the ice flow in the simulations being highly sensitive to tuning the basal traction coefficient, and the

alternate representations of this parameter in the simulations result in the timing of the opening of the Hudson Bay differing by 50–250 years. Accurately representing the model parameters that influence the surface mass balance of the ice sheet (PDD factors and ice topography) and input climatologies (surface air temperature and precipitation) is challenging due to lack of constraining data over the palaeo ice sheet, but they are crucial for the model setup due to the deglaciation being largely driven

by negative surface mass balance (e.g. Carlson et al., 2009).

The agreement of the pattern of deglaciation between simulations and the reconstructed extent (Dyke, 2004) suggests that the model setup can be a useful tool for evaluating the evolution of the early-Holocene Laurentide Ice Sheet with unprecedented model resolution and representation of the ice dynamics. Recent ice sheet reconstructions (Tarasov et al., 2012; Peltier et al., 2015) provide possible deglacial histories for the demise of the Laurentide Ice Sheet, but not in sufficient detail to evaluate the

meltwater flux resulting from particular features of interest, such as the ice saddle collapse over Hudson Bay (Gregoire et al., 2012). The meltwater pulse from this saddle collapse has been hypothesised as having been the primary forcing of the 8.2 ka cooling event (Matero et al., 2017), and these initial simulations reproduce the collapse with a realistic timing between 8.5–8.0 ka (Dyke, 2004; Ullman et al., 2016).

*Code and data availability.*  The input and output data from the simulations described in this paper are available for download from the UK
Polar Data Centre at https://doi.org/10.5285/7E0B2D81-EE71-48D6-A901-3B417D482072 (Matero et al., 2019). The specific versions of the model code used in this study are available for download from the Research Data Leeds repository at https://doi.org/10.5518/778 (Matero et al., 2020).

*Author contributions.*  ISOM and LJG designed the study. ISOM designed, performed and analysed the experiments with inputs from LJG and RFI. ISOM wrote the manuscript with inputs from both co-authors.

*Competing interests.*  The authors have no competing interests to declare.

*Acknowledgements.*  We thank reviewers Michele Petrini and Alexander Robinson for their helpful comments and constructive suggestions. ISOM was funded by the Leeds-York Natural Environment Research Council (NERC) Spheres Doctoral Training Partnership (NE/L002574/1). The contribution from RFI was partly supported by a NERC Independent Research Fellowship (NE/K008536/1). LJG is funded by a UKRI Future Leaders Fellowship (MR/S016961/1). The work made use of the N8 HPC facilities, which were provided and funded by the N8 con-
sortium and EPSRC (EP/K000225/1) and co-ordinated by the Universities of Leeds and Manchester. Modelling support and infrastructure provided within the Faculty of Environment and Centre of Excellence for Modelling the Atmosphere and Climate (CEMAC), University of Leeds. Paul Valdes, RFI and LJG provided the HadCM3 climate simulations. We are grateful to Paul Valdes and Andrew Shepherd for helpful comments on an earlier version of this work, as well as Stephen Cornford and Richard Rigby for helping set up the model.

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
