# Peer review of "Simulating the Early Holocene demise of the Laurentide Ice Sheet with BISICLES (public trunk revision 3298)"

_Geoscientific Model Development, 2019_

## Short Comment (SC1) · 16 Dec 2019

Dear authors,

unfortunately you chose the wrong manuscript type and therefore did not fulfill all requirements for the publication.

The "model experiment description papers" are dedicated to a much more general description of a model experiment design (see http://www.geoscientific-model-development.net/submission/manuscript_types.html ). Your paper is an evaluation specific for one model and one specific model configuration. Therefore it fits into the category for "Development and technical papers".

As such your article needs to fulfill the criteria for a "Development and technical paper", i.e. specifially:

- "Code must be published on a persistent public archive with a unique identifier for the exact model version described in the paper or uploaded to the supplement, unless this is impossible for reasons beyond the control of authors. All papers must include a section, at the end of the paper, entitled "Code availability". Here, either instructions for obtaining the code, or the reasons why the code is not available should be clearly stated. It is preferred for the code to be uploaded as a supplement or to be made available at a data repository with an associated DOI (digital object identifier) for the exact model version described in the paper. Alternatively, for established models, there may be an existing means of accessing the code through a particular system. In this case, there must exist a means of permanently accessing the precise model version described in the paper. In some cases, authors may prefer to put models on their own website, or to act as a point of contact for obtaining the code. Given the impermanence of websites and email addresses, this is not encouraged, and authors should consider improving the availability with a more permanent arrangement. Making code available through personal websites or via email contact to the authors is not sufficient. After the paper is accepted the model archive should be updated to include a link to the GMD paper."

So please provide the information on how to access the model code and ensure the permanent archiving of the published model version.

I will ask the publication office to change the manuscript type of your article.

Best regards,

Astrid Kerkweg

---

## Referee Comment (RC1) · Michele Petrini (Referee) · 16 Jan 2020

General comments:

In this manuscript, the authors use an advanced marine ice sheet model (BISICLES) to simulate the demise of the Laurentide Ice Sheet during the early Holocene (10-7 ka). In particular, the main goal is to simulate a surface mass balance instability event over the ice sheet (known as 'saddle collapse', .ca 8.2 ka) using an ice sheet model with unprecedented horizontal resolution/representation of ice dynamics and ice-ocean in-teraction at the ice sheet marine margins. The simulations' initial conditions are based on the ICE-6G_c reconstruction (ice thickness, topography, bathymetry) and on a previous ice sheet modelling study as concerns the ice temperature. The transient climate forcing is derived interpolating between 500-year intervals climate snapshots simulated with the HadCM3 GCM. The ice sheet sensitivity to different parameters concerning ice dynamics, sub-shelf melting and climate forcing is tested in individual simulations. A simulated deglaciation scenario in agreement with GIA-modelling (ICE_6G_c and GLAC-1d) and empirical reconstructions (Dyke et al. 2004) is presented. The associated meltwater flux magnitude and timing are analysed and compared with estimates based on geological records. Finally, the authors highlight that the ice sheet demise and the associated freshwater fluxes are highly sensitive to the basal traction coefficient and the surface mass balance, with changes in sub-shelf melting and model resolution having a limited effect on timing and duration of the freshwater flux pulse.

The simulations analysed in this manuscript present very advanced modelling features in terms of horizontal resolution (Adaptive Mesh Refinement) and representation of ice dynamics (higher-order approximation of Stokes flow, crevasse calving model). Moreover, this ice sheet model has never been applied before to continental-scale size ice sheets over paleo timescales. For this reason, I think that this manuscript has the potential to provide an essential contribution to the field of paleo-ice sheet modelling.

However, at this point there are some key aspects of this manuscript that need to be reviewed before it can be considered eligible for publication. I think there is need of a deeper analysis to assess the influence of the method used to initialise the simulations on the ice sheet deglaciation and dynamics. Moreover, this manuscript will largely benefit from a more detailed description and analysis of the simulated ice dynamics (evolution of fast-flow areas, calving, grounding-line migration) and the impact of the Adaptive Mesh Refinement on these processes. These aspects represent the main source of innovation of this study and justify the use of a relatively expensive (in terms of computational time) ice sheet model. However, in both results and discussion sections very little/no space is dedicated to this. Therefore, I think this manuscript should be reconsidered after major revisions.

Specific comments:

- All the simulations are initialised starting from (a) ICE-6G_c ice thickness from 10 ka time slice (b) Gregoire et al., 2012 ice temperature from 9 ka time slice, plus 0 °C throughout the ice column outside the ice extent in Gregoire et al., 2012. It is not clear whether the ice velocity is initialised from 0, or a similar approach as for the temperature is used. From the text, it seems that all the simulations are started at 10 ka without previous initialisation of the model thermodynamics. It is remarked in the text that the 2500 simulated years-long 'control' simulation proves that the ice sheet is not in equilibrium with the 10 ka climatic forcing. The choice of not running an initialisation simulation seems to be justified in the text with the sentence "By the early Holocene, the LIS has significantly retreated from its Last Glacial Maximum position and is far from being at equilibrium with the climate". I think that this choice and its implications on the evolution and dynamics of the Laurentide Ice Sheet between 10 and 7 ka deserves a deeper analysis – perhaps to be included in Supplementary Materials. Spin-up simulations are generally 100,000 years-long runs that are used to bring the ice thermodynamics in equilibrium with the climate. Transient simulations of the last deglaciation are generally done starting either from equilibrium-type spin-ups at 21 ka or transient spin-ups starting from the last interglacial (120 ka) and ending at 21 ka (like, for instance, is done in Gregoire et al., 2012). I understand that a similar approach is unfeasible with BISICLES, due to its large computational costs, and this manuscript focus on a shorter time interval (10-7 ka) during which the ice sheet is not in equilibrium with the climate. However, you should still ensure the reader that the method you are using is not producing model artefacts in your simulations due to (1) ice velocity initialisation (2) ice temperatures simulated in Gregoire et al. 2012 had different climate forcing than in this study. It's not fully clear to me whether your climate forcing is the same as in Gregoire et al. 2012 – likely not, as it is PMIP4 – but anyway you take a '9 ka ice temperature' and then you apply a '10 ka' forcing. Also, how areas starting from 0 °C are responding? Do you lose these areas? Perhaps you should include a figure where you show areas with different initialisation for the ice

temperature. I also think you should provide more information on the ice temperature and velocity evolution, both 2D maps and averaged curves. In Figure 3, you show ice velocities at year 50; these velocities does not exhibit clear ice stream patterns, they only seems to be quite high in marine margins - which makes sense, considering how you treat the basal drag coefficient. But is this realistic compared to reconstruction (Margold et al. 2018)? Is this velocity pattern constant or there is a lot of change in fast flow pattern/magnitude? I think that analysing these things in the 'control' simulation might ensure the reader that the deglaciation pattern is 100% caused by changes in your forcing and not because the ice thermodynamics is still in the initialisation phase.

- As stated in the manuscript, using BISICLES allow to simulate the Laurentide Ice Sheet evolution between 10 and 7 ka with high resolution (through the Adaptive Mesh Refinement) and advanced representation of ice dynamics (higher-order approximation of Stokes flow, crevasse calving model). However, I think that the results presented/discussed in this manuscript do not expand enough on this topic, showing mainly ice volume curves and ice thickness maps. I think that aspects of the ice dynamics for which BISICLES is known to be very good (evolution of fast-flow areas, calving, grounding-line migration) should be analysed more and should represent an important (if not central) part of the paper. In the way results are presented, it is hard to understand why it is important/necessary to use BISICLES for this study. The role played by simulated ice dynamics throughout the deglaciation is only assessed indirectly in the sensitivity tests where lower values of the basal traction are considered (with a quite straightforward results). However, there are many ice sheet models less advanced and computationally expensive than BISICLES including a basal drag coefficient – that would likely give the same result in sensitivity tests designed as in this study. Instead, you should try to show why it is really important to use BISICLES for this study, what we are learning from this advanced model. I think for instance that the freshwater flux evolution should be analysed by looking at individual contributions of different processes (sub-shelf melting, calving, runoff). I think you should also include ice velocities maps throughout the deglaciation in the 'standard' simulations (and in the

sensitivity tests, perhaps as supplementary material), showing grounding line migration/ice shelves extent. This could be done for instance for specific snapshot close to the 'saddle collapse' event. Also the Adaptive Mesh Refinement should be discussed more: where, in the grid refinement simulations, the resolution is increased? What are the differences in terms of ice dynamics in these grid points? Looking at the overall volume and freshwater fluxes there is apparently no changes, but how about locally and how about individual processes (sub-shelf melting, calving, runoff)? And if there is still no differences, what we can learn from this experiment? Overall, I would like to see discussed in the paper why the BISICLES simulations performed here tell us more than simulation performed with less advanced ice sheet models. Which were the main dynamical processes occurring during the 'saddle collapse'? If the reader would repeat the same study as in this manuscript with a less advanced model, would he obtain the same results? Why?

Technical corrections:

- Figure 1: it is difficult to identify geographic locations in all the panels, I think you should either add some geographical references to the figures or include a map of the study area. Also adding a map outline would be useful to the reader (like in Figure 3, which is really clear). Moreover, it is difficult to compare panels (a), (b) and (c) as the domain/projection is always different. You could (1) show the three panels in the same ice sheet model domain/projection (2) show only panel (c), adding another panel with geographic locations of the study area. It would be also good to have lat/lon tick marks, instead of native distances.

- Figure 2: in panel (a) caption, you say that colours are yellow, dark blue and green. I can only see yellow, purple and a very tiny portion in light blue/purple. In both panels, it is difficult to identify the geographic locations. You could add a map outline (again, like in Figure 3) and maybe the ICE-6G and GLAC1d ice sheet extent at 10 ka.

- Table 2: you could insert one or multiple last rows with some estimates from geological

records, so that the reader can get an idea of whether individual simulations do a good/bad job in reproducing the peak in freshwater fluxes by looking at this Table.

- Line 17 in Section 4.2: I think you should make more clear (not only here, but in the whole manuscript) when you refer to GIA-modelling reconstructions (ICE6G, GLAC1d) or to fully empirical reconstructions (Dyke 2004, Margold 2018). You use sometimes just 'reconstruction'.

- Line 21 in Section 4.4: you say that the basal melting rate from the geothermal heat flux is negligible and you set it to 0 m/a. This is fine, but how about frictional heating? You should distinguish between geothermal and frictional heat fluxes, otherwise the reader could think that both contributions to basal melting are set to 0 m/a.

- Line 25 in Section 4.4: you should also mention what is happening in terms of oceanic circulation (AMOC strength, warm subsurface Atlantic water export in the North Atlantic) and what are the possible implications for the marine-based sectors of the LIS.

- Line 37 in Section 4.4: it is true that 1 degree resolution in ocean models does not allow to resolve coastlines and shelf cavities. However, simple 2eqs. and 3eqs. sub-shelf melting formulations are forced with far-field ocean temperatures (so, away from the shelf cavities and coastlines). I think it is ok to force your simulations with constant values, but it's not necessarily true that it is a better approach then using transient curves or transient ocean properties based on GCM simulations.

- You could include one or two tables with values for the ice sheet extent/volume at different time slices in 'standard', ICE-6G and GLAC1d, Dyke et al. 2004.

- To increase readability, I suggest to use the same unit for simulated years and dates (in ka). It is difficult to read (for instance) year 1400 and then think that corresponds to 8.6 ka.

- To increase readability, I think the English language needs to be improved.

---

## Referee Comment (RC2) · Alexander Robinson (Referee) · 19 Jan 2020

This manuscript presents a numerical study of the Laurentide deglaciation between 10-8 kyr ago in the region of Hudson Bay, with a specific focus on the 8.2 kyr event. The authors use an innovative approach, by applying the BISICLES ice-sheet model with an adaptive mesh that provides high resolution in dynamic regions. Transient climatic forcing is obtained by interpolating from GCM snapshots through the deglaciation. The experiments successfully reproduce a meltwater pulse with reasonable timing resulting from the separation of two ice domes and increased ablation and ice discharge. I believe that it is quite an interesting study worthy of publication. But I do have trouble

seeing the relevance to GMD. Furthermore, significant revisions are needed to improve the manuscript.

Relevance to journal. I am surprised to see this manuscript submitted to GMD. The main focus of the work seems to be on the scientific results, namely studying the plausibility of the saddle-collapse in Hudson Bay and quantifying rates of FWF and uncertainties. Personally, as it is written now, I would see this paper fitting much better in a journal such as Climate of the Past or even The Cryosphere.

Spin-up time versus experiment. It is clear that significant adjustment occurs in the experiment at the start around 10 kyr. The initialization using ice temperatures from a previous experiment can be seen to have worked reasonably well, as the adjustment seems to complete within just 500 yr or so. Likely this is partly a result of the climatic forcing driving the simulation in the right direction, which is quite interesting in itself. But what is considered part of the experiment itself (that is worth analyzing in terms of volume and distribution, etc.) versus what is just removing inconsistencies due to initialization is not clear from the text and figures. I was surprised to see the detailed figures of just 50 yr of run time, and no figures of the "realistic" ice sheet at, e.g., 8.5 kyr when the authors consider the model to be spun-up. Given the unprecedented resolution of the model, it would be nice to see velocity distributions before during and after the pulse too. I would therefore suggest revising the results section significantly along these lines. Also, note, switching between model years and years is confusing, so I would suggest sticking to just years, which allows comparison with data.

Model resolution. A big emphasis is given to the unparalleled high resolution of the current approach. However, it seems that the authors would have achieved essentially the same results with the $\Omega 0$ setup with no mesh refinement (Fig. 6a). This is touched upon briefly in the text, but the overall feeling from the work is that the authors believe the higher resolution is needed. If this is true, it should be demonstrated more convincingly. For example, I am not convinced that the $\Omega 1$ setup with no isostatic rebound is more realistic than would be an $\Omega 0$ setup with isostatic rebound.

Organization. The Introduction, Discussion and Conclusions are clear and well written. However, the remaining structure of the paper is somewhat hard to follow. In particular, there seems to be some redundancy between the model description and the experimental setup that could be eliminated (e.g., Section 3 versus Section 4.4).

Figures. The quality of figures needs to be improved. Some are not legible (eg, Fig. 9), some contain unexplained features (white rectangles?), etc. Also, for the time series, I would recommend separating the volume and FWF curves into different panels, since these are important curves for the work and difficult to understand as currently plotted.

Please find other more specific comments below.

P1L17: This statement sounds a bit strange: "The new model configuration presented here provides future opportunities to quantify the range of plausible amplitudes and durations of a Hudson Bay ice saddle collapse meltwater pulse and its role in forcing the 8.2 ka event." <= Is this not done in this manuscript directly?

P2L2: Ice Sheet => ice sheet

P2L9: did fully => did not fully

P2L15: kilometre => kilometre-scale [in many places]

P3L6: snow accumulation, ablation => snow accumulation and ablation

P3L26: dynamical => dynamic

P5L15: Define 'r' in flotation equation.

P5L20: The imposed geothermal heat flux sounds more like an experimental setup parameter rather than part of the model description. I would note that this is also already described again later in that section.

Figure 1: The saturated color scale is difficult to see. Consider adding some contours as well. Panels (a) and (b) would perhaps be easier to see if plotted on a projection.

[Figure]

Also, please label the x- and y-axes.

P8L26: I believe the lack of isostatic rebound could be far more important than indicated. Were any tests performed even with a simpler ice sheet model turning on and off isostasy to quantify its impact? Also, it is quite impressive that including isostatic rebound results in 90% computational slowdown – can you explain why?

P8L26: Add UniCiCles reference, or rephrase.

P9L8: my experiment => our experiment

Figure 2: Quality needs to be improved significantly. Color scale in panel (a) makes it difficult to distinguish regions. Can you also explain the border rectangular region? I assume this is outside of the domain of the simulation and therefore should not appear here.

P10L5: inversed => inverted

P11L24: Which range? One value was specified?

P11L32: adn => and

P12L5: First sentence of this paragraph is unnecessary.

P12L7: 'standard' could either be in italics, or in quotes, but both is perhaps not necessary.

P12L15-20: Details of this adjustment do not seem important and could be omitted.

P15L8: Typo "reconsutrction"

P19L2-5: Check units!

Figure 9: Legends too small.

P23L25: than the => than that of the

2019.

GMDD

Interactive
comment

---

## Short Comment (SC2) · 31 Jan 2020

Dear editors,

thank you for changing the manuscript type to the more fitting type.

As suggested in SC1, I am working towards getting a permanently archived copy of the versions of the model code created. I am however unsure as to who to include as the authors of the archived copy. I asked the original authors of the code for advise, and they suggested that I get in touch with GMD about this.

Could you advise me on the best practice on who to include as the authors for the

archived copy? Should it be the authors of this manuscript, with references to the technical reports and journal articles describing the code included in the metadata and abstract. Or alternatively, should the author list on the repository include all the authors of the model codes?

Best regards, Ilkka Matero

—————————————

---

## Author Response (AR1)

**Final author comments in response to reviewer comments by Michele Petrini (RC1) and Alexander Robinson (RC2) on manuscript *"Simulating the Early Holocene demise of the Laurentide Ice Sheet with BISICLES (public trunk revision 3298)"* by Ilkka S. O. Matero et al.**

https://www.geosci-model-dev-discuss.net/gmd-2019-304/#discussion

We thank our reviewers for their positive and constructive comments on our manuscript. Both reviewers raised the following main comments, which have been addressed as outlined in the responses below:

- Initialisation and spin-up requires more discussion. We have updated Figure 3 and will add a new subsection describing the changes in the simulated ice sheet during the "spinup period" (first 1000 years of experiments). This will describe the features and changes that we consider artefacts from the initialisation and will show that our method provides adequate adjustment to initial conditions.
- The role of resolution and ice dynamics needs further discussion. This will be added in sections currently labelled 5.2 and 6.
- Several figures require improvements. We have started making these changes and have included some of our revised figures below.

In addition, Executive Editor Astrid Kerkweg requested that we provide an archive of the model code (SC1). The specific versions of the model code are now archived in the Research Data Leeds repository: https://doi.org/10.5518/778.

Below we provide a point by point response to comments from each reviewer in turn. The comments have been numbered and labelled with (RC1) and (RC2) corresponding to the respective documents uploaded in the online discussion. Our responses are labelled with (AR) and in bold. Edits to the manuscript are labelled with (AE) and in bold. Descriptions of edits done that include specific locations in the manuscript (e.g. P8L5-6 for AE1.3) point to the final version of the manuscript, not the version of the manuscript with tracked changes appended below.

**Comments from Reviewer 1, Michele Petrini**

General comments:
(RC1.1) In this manuscript, the authors use an advanced marine ice sheet model (BISICLES) to simulate the demise of the Laurentide Ice Sheet during the early Holocene (10-7 ka). In particular, the main goal is to simulate a surface mass balance instability event over the ice sheet (known as 'saddle collapse', .ca 8.2 ka) using an ice sheet model with unprecedented horizontal resolution/representation of ice dynamics and ice-ocean interaction at the ice sheet marine margins. The simulations' initial conditions are based on the ICE-6G_c reconstruction (ice thickness, topography, bathymetry) and on a previous ice sheet modelling study as concerns the ice temperature. The transient climate forcing is derived interpolating between 500-year intervals climate snapshots simulated with the HadCM3 GCM. The ice sheet sensitivity to different parameters concerning ice dynamics, sub-shelf melting and climate forcing is tested in individual simulations. A simulated deglaciation scenario in agreement with GIA-modelling (ICE_6G_c and GLAC-1d) and empirical reconstructions (Dyke et al. 2004) is presented. The associated meltwater flux magnitude and timing are analysed and compared with estimates based on geological records. Finally,

the authors highlight that the ice sheet demise and the associated freshwater fluxes are highly sensitive to the basal traction coefficient and the surface mass balance, with changes in sub-shelf melting and model resolution having a limited effect on timing and duration of the freshwater flux pulse. The simulations analysed in this manuscript present very advanced modelling features in terms of horizontal resolution (Adaptive Mesh Refinement) and representation of ice dynamics (higher-order approximation of Stokes flow, crevasse calving model). Moreover, this ice sheet model has never been applied before to continental-scale size ice sheets over paleo timescales. For this reason, I think that this manuscript has the potential to provide an essential contribution to the field of paleo-ice sheet modelling.
**(AR1.1) We thank the reviewer for these positive comments.**

(RC1.2) However, at this point there are some key aspects of this manuscript that need to be reviewed before it can be considered eligible for publication. I think there is need of a deeper analysis to assess the influence of the method used to initialise the simulations on the ice sheet deglaciation and dynamics. Moreover, this manuscript will largely benefit from a more detailed description and analysis of the simulated ice dynamics (evolution of fast-flow areas, calving, grounding-line migration) and the impact of the Adaptive Mesh Refinement on these processes. These aspects represent the main source of innovation of this study and justify the use of a relatively expensive (in terms of computational time) ice sheet model. However, in both results and discussion sections very little/no space is dedicated to this. Therefore, I think this manuscript should be reconsidered after major revisions.
**(AR1.2) We have taken on board these constructive comments and describe below in our response to the specific comments how we plan to revise our manuscript to provide more discussion of initialisation/spinup, and ice dynamics and resolution, including additional figures.**

Specific comments:

(RC1.3) - All the simulations are initialised starting from (a) ICE-6G_c ice thickness from 10 ka time slice (b) Gregoire et al., 2012 ice temperature from 9 ka time slice, plus 0 °C throughout the ice column outside the ice extent in Gregoire et al., 2012. It is not clear whether the ice velocity is initialised from 0, or a similar approach as for the temperature is used.
**(AR1.3) We will edit the manuscript text (current Section 4, describing the model setup) to clarify that ice velocities have been initialized to 0 m a$^{-1}$ everywhere.**
**(AE1.3) P8L5-6: This was added to Section 3 (Model setup and experimental design)**

(RC1.4) From the text, it seems that all the simulations are started at 10 ka without previous initialisation of the model thermodynamics. It is remarked in the text that the 2500 simulated years-long 'control' simulation proves that the ice sheet is not in equilibrium with the 10 ka climatic forcing. The choice of not running an initialisation simulation seems to be justified in the text with the sentence "By the early Holocene, the LIS has significantly retreated from its Last Glacial Maximum position and is far from being at equilibrium with the climate". I think that this choice and its implications on the evolution and dynamics of the Laurentide Ice Sheet between 10 and 7 ka deserves a deeper analysis – perhaps to be included in Supplementary Materials. Spin-up simulations are generally 100,000 years-long runs that are used to bring the ice thermodynamics in equilibrium with the climate. Transient simulations of the last deglaciation are generally done starting either from

equilibrium-type spin-ups at 21 ka or transient spin-ups starting from the last interglacial (120 ka) and ending at 21 ka (like, for instance, is done in Gregoire et al., 2012). I understand that a similar approach is unfeasible with BISICLES, due to its large computational costs, and this manuscript focus on a shorter time interval (10-7 ka) during which the ice sheet is not in equilibrium with the climate. However, you should still ensure the reader that the method you are using is not producing model artefacts in your simulations due to (1) ice velocity initialisation (2) ice temperatures simulated in Gregoire et al. 2012 had different climate forcing than in this study.

**(AR1.4) We will extend the analysis of the adjustments to initial conditions in our manuscript. However, we prefer to keep this within the main manuscript rather than putting it in the supplementary materials. We have decided to label the first 1000 years of the simulations as the "Spin-up phase" and we will describe changes in ice velocity, temperature and surface elevation during this phase in greater detail, accompanied with new figures (see comments below).**

**(AE1.4) We have added (i) a new section describing the changes in the ice sheet during the spin-up phase (Section 4.1), (ii) updated Figure 3 showing the differences in depth-integrated ice velocity magnitudes and (iii) updated Figure 2 showing the ice temperature initialisation.**

(RC1.5) It's not fully clear to me whether your climate forcing is the same as in Gregoire et al. 2012 – likely not, as it is PMIP4 – but anyway you take a '9 ka ice temperature' and then you apply a '10 ka' forcing.

**(AR1.5) Correct, the climate forcing used to drive these experiments is not the same as in Gregoire et al. (2012) but uses more up to date climate simulations as described in P6L17-28. This will be clarified in the manuscript.**

**(AE1.5) P6L18-26: We added information on the simulations being based on a similar, but updated methodology as Gregoire et al., (2012).**

(RC1.6) Also, how areas starting from 0 °C are responding? Do you lose these areas? Perhaps you should include a figure where you show areas with different initialisation for the ice temperature. I also think you should provide more information on the ice temperature and velocity evolution, both 2D maps and averaged curves.

**(AR1.6) Initialising temperatures to 0 °C in some regions has the effect of promoting ice flow (through eq. 3). However, the areas where ice temperature was initialised at 0 °C degrees are also initialised with thin ice from the ICE-6G_c reconstruction. Thus, ice in these regions retreats within the first few hundred years of simulations because ice either reaches floatation point and eventually melts or calves or low surface elevation leads to high melt rates. We provide a Revised Figure 2 (see below) showing initial ice thickness and temperature to illustrate this. Unfortunately, the ice temperature output of the simulations was not saved as part of the archived data, and are no longer available. This work was done as part of a PhD study that finished in 2018 and we do not have any resources left for rerunning the simulations. Thus, we are unable to produce maps of ice sheet temperature at different times during the simulations. We have, however, produced new 2D maps of ice velocity and will expand the discussion of the effect of ice temperature initialisation in our manuscript, as detailed in our response to the next point below.**

**(AE.1.6) Revised Figures 2 and 3 (below) are now included in the manuscript. P13L5-L19: We have expanded on the discussion of the ice temperature initialisation in Section 4.1 (LIS evolution during the initial adjustment period).**

[Figure]

*Revised Figure 2: **(a)** Initial ice thickness (based on ICE-6G_c 10 ka time slice; Peltier et al., 2015), with the coastline shown in black. **(b)** Initial ice temperature at the lowest level of the ice sheet (based on 9 ka time slice of the simulation by Gregoire et al., 2012).*

(RC1.7) In Figure 3, you show ice velocities at year 50; these velocities does not exhibit clear ice stream patterns, they only seems to be quite high in marine margins - which makes sense, considering how you treat the basal drag coefficient. But is this realistic compared to reconstruction (Margold et al. 2018)? Is this velocity pattern constant or there is a lot of change in fast flow pattern/magnitude? I think that analysing these things in the 'control' simulation might ensure the reader that the deglaciation pattern is 100% caused by changes in your forcing and not because the ice thermodynamics is still in the initialisation phase.

**(AR1.7) We have prepared a revised version of Figure 3 (see below) that shows modelled ice velocity at 9 ka, at 8.5 ka and 8.0 ka compared with the ice stream reconstruction of Margold et al., (2018; panels taken from their Figure 5). A paragraph discussing the ice dynamics in relation to the ice stream locations will be added to section 5.1 (Laurentide Ice Sheet evolution in 'standard' and 'control' simulations). We find that the locations of our main ice streams match at least four reconstructed major ice streams (Hudson Strait, Ungawa Bay, James Bay and the Hayes Lobe streams), but in general the ice streams are quite broad, particularly over Hudson Bay and surrounding the Labrador Dome. The new panels in Figure 3 show that after 50 years of simulations, artificial patterns in ice velocity are present as a result of adjustments to the initial ice geometry, because the initial ice sheet surface is incompatible with ice dynamics. However, these artefacts disappear within the first 1000 years of simulation as the ice is redistributed by BISICLES. In response to these comments, we plan to add a new section to the manuscript describing the initial adjustment and changes during the spinup phase, expanding on our analysis of the evolution of ice topography, extent and velocity and how these relate to the experimental design.**

**(AE1.7) The updated figure 3 and P13L21-P15L5: discussion of ice dynamics has been added as outlined in AR1.7 above to Section 4.2 (Laurentide Ice Sheet evolution in 'standard' and 'control' simulations). As stated in AE1.6, we have included the new section describing the initial adjustment phase and how these relate to the experimental design.**

[Figure]

*Revised Figure 3: Reconstruction of LIS ice stream activity at 10.1 ka **(a)** and 8.9 ka **(d)** (adapted from fig 5. in Margold et al., 2018). The ice streams that were active at the respective times are shown in light blue and numbered in black, ice streams that 'switched off' in the preceding 1000 years are shown in grey and those that 'switched on' shown in dark blue with numbering in red. Panels **(b)**, **(c)**, **(e)** and **(f)** show the magnitude (m a⁻¹) of depth-integrated velocities in the 'standard' simulation at 9.95 ka, 9.00 ka, 8.5 ka and at the end of the simulation (8.00 ka) respectively (note, Greenland is not included in the simulations). Coastlines are shown with the black contour in the panels showing the simulated ice velocities.*

(RC1.8) - As stated in the manuscript, using BISICLES allow to simulate the Laurentide Ice Sheet evolution between 10 and 7 ka with high resolution (through the Adaptive Mesh Refinement) and advanced representation of ice dynamics (higher-order approximation of Stokes flow, crevasse calving model). However, I think that the results presented/discussed in this manuscript do not expand enough on this topic, showing mainly ice volume curves and ice thickness maps. I think that aspects of the ice dynamics for which BISICLES is known to be very good (evolution of fast-flow areas, calving, grounding-line migration) should be analysed more and should represent an important (if not central) part of the paper. In the way results are presented, it is hard to understand why it is important/necessary to use BISICLES for this study. The role played by simulated ice dynamics throughout the deglaciation is only assessed indirectly in the sensitivity tests where lower values of the basal traction are considered (with a quite straightforward results). However, there are many ice sheet models less advanced and computationally expensive than BISICLES including a basal drag coefficient – that would likely give the same result in sensitivity tests designed as in this study. Instead, you should try to show why it is really important to use BISICLES for this study, what we are learning from this advanced model. I think for instance that the freshwater flux evolution should be analysed by looking at individual contributions of different processes (sub-shelf melting, calving, runoff).

**(AR1.8) We appreciate that the reviewer is interested in the role of ice dynamics and the capabilities of the BISICLES model in the context of our work. We will extend the text to**

include more discussion of how the representation of ice sheet dynamics influences ice evolution in this context, answering some of the specific points raised here (see responses below). But first, we wish to remind the reviewer and reiterate to the editor that this is a 'Technical Manuscript' for GMD (see response to RC2.1) and is not intended to analyse in detail the processes involved in the Hudson Bay Saddle collapse. Therefore, an in-depth analysis of the contribution to the freshwater flux from individual processes is not appropriate, falling beyond the scope of this study. It will instead form the basis of future work, for which this technical manuscript is the first step.

**(AE1.8) We have added the revised figure 3 (above) and included a paragraph (P13L21-P15L5) discussing the importance of representation of ice dynamics and comparison with ice stream reconstructions. We have also expanded on the discussion of ice dynamics and the mesh refinement by adding two paragraphs to the discussion section (P26L8-P26L32).**

(RC1.9) I think you should also include ice velocities maps throughout the deglaciation in the 'standard' simulations (and in the sensitivity tests, perhaps as supplementary material), showing grounding line migration/ice shelves extent. This could be done for instance for specific snapshot close to the 'saddle collapse' event.

**(AR1.9) A new figure (the one below after further refinement) showing the ice velocity and grounding line position in the standard simulation and simulations with different levels of resolution will be added to the manuscript.**

**(AE1.9) The ice velocity maps showing the integrated ice velocity in 'standard' at 4 different times are now included in the revised figure 3 and prior to the saddle collapse in figure 7. The simulation timings to be shown in the panels were chosen along the suggestions in this comment and RC2.3 and RC2.4. Figure 7 also shows the suggested extent of the ice shelves prior to the saddle collapse, as well as the impact of increasing the AMR level from 1 to 2 on the grounding line and the ice shelves on both sides of the saddle.**

[Figure]

*Figure 7: Magnitude of vertically integrated ice velocity at model year 1700 with the grounding line and coastlines shown in black. (a) The velocity over the whole domain in the 'standard' simulation. A zoomed-in view of the ice saddle (b) in 'standard', and (c) in 'AMR_2'.*

(RC1.10) Also the Adaptive Mesh Refinement should be discussed more: where, in the grid refinement simulations, the resolution is increased? What are the differences in terms of ice dynamics in these grid points? Looking at the overall volume and freshwater fluxes there is apparently no changes, but how about locally and how about individual processes (sub-shelf melting, calving, runoff)? And if there is still no differences, what we can learn from this

experiment? Overall, I would like to see discussed in the paper why the BISICLES simulations performed here tell us more than simulation performed with less advanced ice sheet models. Which were the main dynamical processes occurring during the 'saddle collapse'? If the reader would repeat the same study as in this manuscript with a less advanced model, would he obtain the same results? Why?

**(AR1.10) We will expand the discussion on the impact of resolution and the adaptive mesh refinement level in the section 5.2 (Impact of varying the mesh resolution) and in the Discussion. As already briefly highlighted in section 5.2 and section 5.5, increasing the resolution does affect the timing of the peak FWF and the grounding line migration, and we will expand on this. To give more thorough context, we will also discuss that the low resolution (10 km) is sufficient for making a first order evaluation of the meltwater flux associated with the Hudson Bay Saddle Collapse. This is because the event is primarily driven by a surface mass balance feedback and is therefore not so affected by resolution at the grounding line, for example. That said, simulating a realistic meltwater flux does rely on producing a realistic simulation of surface elevation, which highly depends on ice dynamics. Thus, a good representation of ice streams through membrane stresses (as in BISICLES) may be important in adequately representing this event. We will incorporate all of these topics in our expanded section 5.2 and Discussion. To answer the interesting questions raised by the reviewer more definitely, a multi model intercomparison exercise involving ice sheet models of different complexities would be needed.**

**(AE1.10) P17L14-33: We have modified section 4.3 (Impact of varying the mesh refinement level) in line with our response above and added figure 7 to clarify the points made in the text. We have expanded the discussion of importance of applying the AMR and high resolution to discussion (P26L23-32)**

Technical corrections:

(RC1.11) - Figure 1: it is difficult to identify geographic locations in all the panels, I think you should either add some geographical references to the figures or include a map of the study area. Also adding a map outline would be useful to the reader (like in Figure 3, which is really clear). Moreover, it is difficult to compare panels (a), (b) and (c) as the domain/projection is always different. You could (1) show the three panels in the same ice sheet model domain/projection (2) show only panel (c), adding another panel with geographic locations of the study area. It would be also good to have lat/lon tick marks, instead of native distances.

**(AR1.11) We have created a Revised Figure 1 (see below) showing only panel c from the previous Figure 1 and plotted in the same style as current Figure 3, which the reviewer thought was very clear. We will also superimpose graticules (i.e. latitude longitude lines) onto the new panels in Figure 1.**

**(AE1.11) Revised figure 1 is now included in the manuscript. We have also added the graticules to other figures visualising the ice sheet apart from new Figure 7. Reasoning for this is that for these figures the 10 km -grid makes it easier to compare the differences between the panels.**

[Figure]

*Revised Figure 1:* **(a)** *The 'ETOPO 10 ka' topography map centred at Hudson Bay on a 10 km grid. The basal topography in the northern part of the grid (Greenland and Canadian archipelago) is set to -1000 m to avoid ice sheet formation in these regions. The black lines show contours every 400 m.* **(b)** *Basal traction coefficient map used in the 'standard' simulation. The basal traction coefficient* **C** *values are calculated based on bedrock elevation and sediment coverage. The domain was divided into three different types of sediment coverage: bare bedrock, sediment-covered and submerged bare bedrock in the Hudson Strait (highlighted in blue in panel (b), C = 80 Pa m$^{-1}$ a). The outermost 35 grid cells along both axes are initialised with a high basal traction coefficient (C = 450 Pa m$^{-1}$ a) to avoid ice flow reaching the edges of the domain.*

(RC1.12) - Figure 2: in panel (a) caption, you say that colours are yellow, dark blue and green. I can only see yellow, purple and a very tiny portion in light blue/purple. In both panels, it is difficult to identify the geographic locations. You could add a map outline (again, like in Figure 3) and maybe the ICE-6G and GLAC1d ice sheet extent at 10 ka.

**(AR1.12) The panel from Figure 2 has been replotted with a different colour scheme and included as panel b in Revised Figure 1 (see above). A Revised Figure 2 now has a panel showing a map of the ICE-6G_c ice thickness at 10 ka with coast lines plotted in the same style as Figure 3.**

**(AE1.12) Revised figures 1 and 2 are now included in the manuscript. The panel from Figure 2 has been replotted with a different colour scheme and included as panel b in Revised Figure 1 (see above). A Revised Figure 2 now has a panel showing a map of the ICE-6G_c ice thickness at 10 ka with coast lines plotted in the same style as Figure 3.**

(RC1.13) - Table 2: you could insert one or multiple last rows with some estimates from geological records, so that the reader can get an idea of whether individual simulations do a good/bad job in reproducing the peak in freshwater fluxes by looking at this Table.

**(AR1.13) There is unfortunately no direct geological evidence that constrain the duration and amplitude of the peak of freshwater. Records of freshening of the Labrador Sea are difficult to interpret and have large age uncertainties. Lines of reasoning combining climate proxy records and model outputs could be used, but this would require substantial work to review the evidence available. There are also sea level records that could be used to derive estimates of volume of sea level rise, but uncertainties in the records, contributions from background melt of other ice sheets and isostatic sea level changes**

complicate the interpretation of the records. We are working on this in the long run, but such work is not possible at this time and falls outside the scope of this study, which focuses on the technical considerations of modelling the Hudson Bay Saddle collapse with the BISICLES model.

**(AE1.13) No edits done in response to this comment.**

(RC1.14) - Line 17 in Section 4.2: I think you should make more clear (not only here, but in the whole manuscript) when you refer to GIA-modelling reconstructions (ICE6G, GLAC1d) or to fully empirical reconstructions (Dyke 2004, Margold 2018). You use sometimes just 'reconstruction'.

**(AR1.14) This will be clarified in the manuscript.**

**(AE1.14) This has been clarified in the manuscript in places where there were unclear references to reconstructions, e.g.: P10L7, P10L11 and P17L4.**

(RC1.15) - Line 21 in Section 4.4: you say that the basal melting rate from the geothermal heat flux is negligible and you set it to 0 m/a. This is fine, but how about frictional heating? You should distinguish between geothermal and frictional heat fluxes, otherwise the reader could think that both contributions to basal melting are set to 0 m/a.

**(A1.15) Indeed, frictional heating induces basal melt in our simulations. We will revise this sentence to clarify this.**

**(AE1.15) P11L11-13: We have clarified that neither frictional or basal heat flux result in basal melt in this version of BISICLES, but that these affect the ice flow through affecting the temperature, which in part determines the effective viscosity at the base. The text has been modified to the following: "While basal and frictional heat fluxes do not directly result in ice melt at the base in these simulations, they do affect the ice flow through changing the effective viscosity at the base of the ice sheet as discussed in Section 2.1."**

(RC1.16) - Line 25 in Section 4.4: you should also mention what is happening in terms of oceanic circulation (AMOC strength, warm subsurface Atlantic water export in the North Atlantic) and what are the possible implications for the marine-based sectors of the LIS.

**(AR1.16) We will add a sentence on the changes on ocean circulation within the period of interest and how it might have affected the LIS. Even at present day it is difficult to relate changes in ocean circulation to sub-shelf melt rates. We therefore chose to assess the model sensitivity to a large range of values for the sub-shelf melt rates (2-45 m/a), to which the model setup showed little sensitivity (see section 5.5, Impact of varying the sub-shelf melting rate).**

**(AE1.16) P23L6-L18: We have expanded the discussion of further model development related to the sub-shelf melting rate in Section 4.6 (Impact of varying the sub-shelf melt rate), to include the potential minor impact of changes in the AMOC and the associated heat transport.**

(RC1.17) - Line 37 in Section 4.4: it is true that 1 degree resolution in ocean models does not allow to resolve coastlines and shelf cavities. However, simple 2eqs. and 3eqs. subshelf melting formulations are forced with far-field ocean temperatures (so, away from the shelf cavities and coastlines). I think it is ok to force your simulations with constant values, but it's not necessarily true that it is a better approach then using transient curves or transient ocean properties based on GCM simulations.

**(AR1.17)** Existing equations that relate sub-shelf melt to ocean temperature (often off the shelf) are simple and need to be calibrated for specific basins in Antarctica and Greenland for example. Whether such equations would capture the circulation within the Hudson Straight when it was influenced by marine terminating glaciers is unclear. We expect that glacier melt would have strongly influenced the density, temperature and circulation of waters within the straight, making it difficult to relate sub-shelf melt within the straight to ocean temperatures in the Labrador Sea. This is why we chose to simply represent sub-shelf melt as constant through time. We will clarify our reasoning in section 4.4 of the manuscript.

**(AE1.17)** P11L4-L26: We have separated the discussion of relating the sub-shelf melt rate with sea surface temperatures into its own paragraph to subsection 3.4 (Basal fluxes).

(RC1.18) - You could include one or two tables with values for the ice sheet extent/volume at different time slices in 'standard', ICE-6G and GLAC1d, Dyke et al. 2004.

**(AR1.18)** The comparison of ice sheet volume with ICE-6G_c and GLAC-1D is already available in Figures 4 and 6. Since we do not intend to validate our results against observations (it is not the purpose of the manuscript), the tables suggested above would be redundant. We feel this would crowd the manuscript, deviating from the Technical Manuscript category that it is submitted under, and therefore prefer not to add them.

**(AE1.18)** No edits done to the manuscript in response to this comment.

(RC1.19) - To increase readability, I suggest to use the same unit for simulated years and dates (in ka). It is difficult to read (for instance) year 1400 and then think that corresponds to 8.6 ka.

**(AR1.19)** We will follow this suggestion.

**(AE1.19)** We have followed this suggestion.

(RC1.20) - To increase readability, I think the English language needs to be improved.

**(AR1.20)** Reviewer 2 particularly highlighted that the manuscript is 'well written' and we agree with this general assessment, noting that we will follow Reviewer 2's suggested structural edits (RC2.7). We have spotted and will also correct a few typographical errors in the manuscript to improve the readability in those places.

**(AE1.20)** We have corrected a few typographical errors, see our responses to RC2.12, RC2.13, RC2.14, RC2.24, RC2.28 for examples of specific edits. Other very minor edits have been made to correct spelling and grammar errors; see tracked changes version of the article to see where.

**Comments from Reviewer 2, Alexander Robinson (RC2)**

**General comments:**

(RC2.1) This manuscript presents a numerical study of the Laurentide deglaciation between 10- 8 kyr ago in the region of Hudson Bay, with a specific focus on the 8.2 kyr event. The authors use an innovative approach, by applying the BISICLES ice-sheet model with an adaptive mesh that provides high resolution in dynamic regions. Transient climatic forcing is obtained by interpolating from GCM snapshots through the deglaciation. The experiments successfully reproduce a meltwater pulse with reasonable timing resulting from the separation of two ice domes and increased ablation and ice discharge. I believe that it is

quite an interesting study worthy of publication. But I do have trouble seeing the relevance to GMD. Furthermore, significant revisions are needed to improve the manuscript.

**(AR2.1) Thank you for the constructive feedback and comments. Please see answers to specific comments below including the relevance to GMD.**

(RC2.2) Relevance to journal. I am surprised to see this manuscript submitted to GMD. The main focus of the work seems to be on the scientific results, namely studying the plausibility of the saddle-collapse in Hudson Bay and quantifying rates of FWF and uncertainties. Personally, as it is written now, I would see this paper fitting much better in a journal such as Climate of the Past or even The Cryosphere.

**(AR2.2) We disagree. This is a Technical Manuscript that specifically pertains to describing how the BISICLES ice sheet model is used in a new palaeo setting (including the sensitivity of uncertain parameter choices, usually calibrated to directly observed ice behaviour, in this context). This is initial, technical work to setup further study of the 8.2 kyr event and we do not propose that we have found the scientific answers to explain the event. This fits within the category of "Development and technical papers" as highlighted by the executive editor comment SC1 in the interactive discussion. The sensitivity study and initial results regarding the rates and timing of FWF release demonstrate the usefulness of the model setup for studying the Hudson Bay Saddle collapse and 8.2 kyr event. Understanding the sensitivity of the results to choices in modelling setup for such novel, palaeo applications is an important technical step for eventually being able to reach a better understanding of the events through further modelling work (e.g. to quantify the range of plausible rates of meltwater fluxes). Such additional investigation that builds on the work presented here would, we hope, constitute an excellent Climate of the Past or Cryosphere type manuscript. However, in the meantime, it exceeds the purpose of the present study. We will edit the Introduction to clarify the technical framing of the work and emphasise the fit to GMD 'development and technical papers'.**

**(AE2.2) P3L34-P4L3: We have clarified this in the last part of the introduction as follows: "This technical manuscript presents the model configuration and its sensitivity to the aforementioned model parameters. Understanding the sensitivity of the results to choices in modelling setup for such novel, palaeo applications is an important technical step for eventually being able to reach a better understanding of the recorded events through further modelling work; for example, to quantify the range of plausible rates of meltwater fluxes, which would then provide a powerful (hitherto elusive) test for climate model performance (Schmidt and Legrande, 2005, DOI: 10.1016/j.quascirev.2005.01.015)."**

(RC2.3) Spin-up time versus experiment. It is clear that significant adjustment occurs in the experiment at the start around 10 kyr. The initialization using ice temperatures from a previous experiment can be seen to have worked reasonably well, as the adjustment seems to complete within just 500 yr or so. Likely this is partly a result of the climatic forcing driving the simulation in the right direction, which is quite interesting in itself. But what is considered part of the experiment itself (that is worth analyzing in terms of volume and distribution, etc.) versus what is just removing inconsistencies due to initialization is not clear from the text and figures. I was surprised to see the detailed figures of just 50 yr of run time, and no figures of the "realistic" ice sheet at, e.g., 8.5 kyr when the authors consider the model to be spun-up.

**(AR2.3) We note that this comment is in line with a comment from Reviewer 1 who requested a more in-depth analysis of the effects of initialization (RC1.4). We will add a new**

section to the manuscript describing these results. To clarify the evaluation of these results, we have labelled the first thousand years of the simulation (10-9 ka) the spin-up phase and will describe the adjustments in terms of ice surface, extent and velocity during this phase and the subsequent transient-experiment phase. As stated above (see response to RC1.7) our revised Figure 3 provides further plots of the ice sheet after the spin-up phase.

**(AE2.3) We have added the revised Figure 3 showing plots of the ice sheet and its velocity distribution at 9.95, 9.0, 8.5 and 8.0 ka. A new section 4.1 ('LIS evolution during the initial adjustment period') has been added to discuss the evolution of the ice sheet over the first 1000 years.**

(RC2.4) Given the unprecedented resolution of the model, it would be nice to see velocity distributions before during and after the pulse too. I would therefore suggest revising the results section significantly along these lines.

**(AR2.4) The new Figure 3 shows ice velocity at 9 ka, at 8.5 ka and 8.0 ka as suggested and we plan to add a discussion of ice velocity and a comparison to proxy data in the subsection discussing the ice sheet evolution in the 'standard' simulation (currently Section 5.1). After further revision, we will include a new figure at 8.3 ka showing the integrated ice velocity and grounding line location with different levels of mesh refinement. Draft of the figure is currently under comment RC1.9.**

**(AE2.4) In addition to the two new figures (Figures 3 and 7) that combined show the velocity distribution at 9 ka, 8.5 ka, 8.3ka and 8.0 ka, we have added a new paragraph P13L21-P15L5 discussing the ice velocities and a comparison with a reconstruction of ice stream locations. See also our response to another comment in line with this one (RC 1.8).**

(RC2.5) Also, note, switching between model years and years is confusing, so I would suggest sticking to just years, which allows comparison with data.

**(AR2.5) This was also suggested by Reviewer 1 (RC1.19). As suggested, we will change the text from referring to model years to the respective timing in ka when discussing specific times in the simulation (see also our response to RC1.19).**

**(AE2.5) We have changed the references from model years to respective timing in ka.**

(RC2.6) Model resolution. A big emphasis is given to the unparalleled high resolution of the current approach. However, it seems that the authors would have achieved essentially the same results with the $\Omega 0$ setup with no mesh refinement (Fig. 6a). This is touched upon briefly in the text, but the overall feeling from the work is that the authors believe the higher resolution is needed. If this is true, it should be demonstrated more convincingly. For example, I am not convinced that the $\Omega 1$ setup with no isostatic rebound is more realistic than would be an $\Omega 0$ setup with isostatic rebound.

**(AR2.6) This comment discusses similar concerns to comment RC1.10 by reviewer 1, and will be addressed by expanding the discussion of the impact of the resolution and adaptive mesh refinement level in former sections 5.5 and 6 (AR1.10). The key message is that the low resolution in these simulations (10 km grid size) is sufficient for the large-scale evolution of the ice sheet, but the individual processes such as ice streams, the grounding line migration and timing of the peak of the FWF would benefit from further refinement of the mesh due to e.g. the diverging peak FWF timing between AMR levels 0&1 and 1&2.**

**(AE2.6) We have added figure 7 (also shown in this document under RC1.9) to clarify our points made in the text (P17L27-31 and P26L23-32), discussing the importance of the**

**higher resolution for detailed quantification of the saddle collapse and resulting meltwater pulse, as opposed to providing a first-order estimate with the low resolution.**

(RC2.7) Organization. The Introduction, Discussion and Conclusions are clear and well written. However, the remaining structure of the paper is somewhat hard to follow. In particular, there seems to be some redundancy between the model description and the experimental setup that could be eliminated (e.g., Section 3 versus Section 4.4).

**(AR2.7) Thank you for pointing this out. Section three originally included a longer description of the surface mass balance and how it is implemented in the model setup. Contents from current section 3 will be moved under the model description (to create a new subsection 2.4). The current subsection 4.4 will be split between two updated subsections describing basal fluxes (3.4) and Surface mass balance (3.5).**

**(AE2.7) We moved the contents from current section 3 under the model description (to a new subsection 2.4). We split the current subsection 4.4 between two updated subsections describing basal fluxes (3.4) and Surface mass balance (3.5).**

(RC2.8) Figures. The quality of figures needs to be improved. Some are not legible (eg, Fig. 9), some contain unexplained features (white rectangles?), etc. Also, for the time series, I would recommend separating the volume and FWF curves into different panels, since these are important curves for the work and difficult to understand as currently plotted.

**(AR2.8) We will improve all the figures in the manuscript as guided (e.g. see Revised Figures 1-3). Legends in Fig. 9 will be enlarged and moved to the outer edge of the figure. Key panels from Figures 1 and 2 have been combined and redrawn according to the suggested edits from both reviewers (see Revised Figures 1 and 2). The white rectangles outside of the ice sheet area in Figure 1 underlie the figure legends so that they are readable, but they do not obscure the plot. Figure 3 has been redrawn to show the integrated ice velocities at different time steps as suggested. We will clarify the time series figure by removing the first 500 years of the simulation and separating the evolution of the ice sheet volume and the freshwater flux into individual panels for each parameter.**

**(AE2.8) Figures 1-3 and 9 (currently Figure 10) have been modified. Graticules have been added to figures 1, 2, 3, 5, 8, 9 and 10. The resolution of Figures 8 and 9 has been improved and clear coastlines have been added to both figures and figure 10. We have not updated the timeseries plot (fig. 6) because we feel that the figure is actually easier to interpret with both the volumes and freshwater flux curves in the same figure.**

**Specific comments**:
(RC2.9) P1L17: This statement sounds a bit strange: "The new model configuration presented here provides future opportunities to quantify the range of plausible amplitudes and durations of a Hudson Bay ice saddle collapse meltwater pulse and its role in forcing the 8.2 ka event." <= Is this not done in this manuscript directly?

**(AR2.9) This manuscript presents the model configuration and the key sensitivities of this model setup to varying different model parameters. While we do present the amplitude and the duration of the meltwater pulse in our set of simulations, further development of the experimental setup and further comparisons with observations are needed to reliably quantify these. This framing will be clarified in the revised manuscript (also see our responses to RC1.8, RC1.13 and RC2.2, above).**

**(AE2.9) P3L34-P4L3: We clarified this in the last paragraph of the introduction: 'This technical manuscript presents the model setup and its sensitivity to the aforementioned**

model parameters. Understanding the sensitivity of the results to choices in modelling setup for such novel, palaeo applications is an important technical step for eventually being able to reach a better understanding of the events through further modelling work (e.g. to quantify the range of plausible rates of meltwater fluxes).'

(RC2.10) P2L2: Ice Sheet => ice sheet
(RC2.11) P2L9: did fully => did not fully
(RC2.12) P2L15: kilometre => kilometre-scale [in many places]
(RC2.13) P3L6: snow accumulation, ablation => snow accumulation and ablation
(RC2.14) P3L26: dynamical => dynamic
(RC2.15) P5L15: Define 'r' in flotation equation.
**(AR2.10-15) We will implement these changes. Thank you for pointing these out.**
**(AE2.10) P2L2: Changed as suggested.**
**(AE2.11) P2L9: Changed as suggested.**
**(AE2.12) P1L9, P2L15, P4L11&P4L12, P26L29: Changed as suggested.**
**(AE2.13) P3L6: Changed as suggested.**
**(AE2.14) P2L27, P3L7, P3L26, P3L7&L16, P15L7, P23L7, P26L10+L14: Changed as suggested.**
**(AE2.15) P5L18: 'r' has been defined: "where r is bedrock elevation".**

(RC2.16) P5L20: The imposed geothermal heat flux sounds more like an experimental setup parameter rather than part of the model description. I would note that this is also already described again later in that section.
**(AR2.16) The description of the imposed flux will be removed from the text here, as it is indeed already described in 4.4 and fits better under the experimental setup.**
**(AE2.16) P11L4-L13: The description of the imposed flux has been removed from the model description and is now in subsection 3.4 (Basal fluxes) in the experiment design section.**

(RC2.17) Figure 1: The saturated color scale is difficult to see. Consider adding some contours as well. Panels (a) and (b) would perhaps be easier to see if plotted on a projection. Also, please label the x- and y-axes.
**(AR2.17) We have done this in our Revised Figure 1.**
**(AE2.17) Revised figure 1 is now included in the manuscript.**

(RC2.18) P8L26: I believe the lack of isostatic rebound could be far more important than indicated. Were any tests performed even with a simpler ice sheet model turning on and off isostasy to quantify its impact? Also, it is quite impressive that including isostatic rebound results in 90% computational slowdown – can you explain why?
**(AR2.18) We did not perform tests with a simpler ice sheet model with isostasy. The slowdown was due largely to technical inefficiency of the code. A newer version of the isostatic module is currently under development to solve this.**
**(AE2.18) No edits were done to the manuscript in response to this comment.**

(RC2.19) P8L26: Add UniCiCles reference, or rephrase.
**(AR2.19) We will remove the reference to 'UniCiCles' which is confusing here. UniCiCles refers to the coupling code between the UKESM and BISICLES, but the part that relates to isostatic adjustment is unpublished and untested. Instead we will revise the sentence to mention other ongoing code developments to include a Glacial Isostatic adjustment model within the BISICLES model.**

**(AE2.19) P8L32-33: We have removed the reference to 'UniCiCles' from the text and revised the sentence to discuss ongoing code development within the BISICLES model.**

(RC2.20) P9L8: my experiment => our experiment
**(AR2.20) We will correct this in the revised manuscript.**
**(AE2.20) P10L3: We have corrected this in the revised manuscript.**

(RC2.21) Figure 2: Quality needs to be improved significantly. Color scale in panel (a) makes it difficult to distinguish regions. Can you also explain the border rectangular region? I assume this is outside of the domain of the simulation and therefore should not appear here.
**(AR2.21) We have done this in Revised Figures 1 and 2 by merging panel (b) from Fig. 2 with Panel (c) from Fig. 1. The colour scale has been changed to black and white with discreet changes every 100 (Pa m$^{-1}$ a). An explanation of the border region has been added to the caption. The outermost 35 cells along each axis were set to the bedrock value for the basal traction coefficient to prevent ice flow to the edges of the domain, which could have caused instabilities. The border regions are outside of the ice sheet, and this change does not affect the dynamics of the LIS.**
**(AE2.21) Revised figures 1 and 2 are now included in the manuscript, with the changes described in our response above (AR2.21).**

(RC2.22) P10L5: inversed => inverted
**(AR2.22) We will correct this in the revised manuscript.**
**(AE2.22) P11L1: We have corrected this in the revised manuscript.**

(RC2.23) P11L24: Which range? One value was specified?
**(AR2.23) This referred to the range of contemporary sub-shelf melt rates observed at Antarctic ice shelves. We will change the wording at the start of the paragraph to not refer to a range of values, but to discuss the value itself and reasons for choosing it.**
**"A value of 15 m a$^{-1}$ was chosen for the sub-shelf melt rate in the 'standard' simulation. The orbital configuration of the Earth approaching the Holocene Climatic Optimum conditions post 10 ka (Kaufman et al., 2004), and the associated increase in radiative forcing could have resulted in substantial heat absorption to the lake and sea water adjacent to the ice sheet. The chosen value is comparable to areas of high sub-shelf melt in the modern-day Antarctic Ice Sheet, where the rates have been estimated to be up to 43 m a$^{-1}$ (Rignot et al., 2013)."**
**(AE2.23) P11L14-16: We have updated the text as the following: "A value of 15 m a$^{-1}$ was chosen for the sub-shelf melt rate in the 'standard' simulation. The chosen value is comparable to areas of high sub-shelf melt in the modern-day Antarctic Ice Sheet, where the rates have been estimated to be up to 43 m a$^{-1}$ (Rignot et al., 2013)." See also our response to RC1.16.**

(RC2.24) P11L32: adn => and
**(AR2.24) We will correct this in the revised manuscript.**
**(AR2.24) The sentence has been removed from the manuscript.**

(RC2.25) P12L5: First sentence of this paragraph is unnecessary.
**(AR2.25) We will remove this sentence.**

**(AE2.25) P12L19: We have removed the sentence.**

(RC2.26) P12L7: 'standard' could either be in italics, or in quotes, but both is perhaps not necessary.
**(AR2.26) We will correct this throughout the manuscript and ensure consistency.**
**(AE2.26) This has been corrected throughout the manuscript.**

(RC2.27) P12L15-20: Details of this adjustment do not seem important and could be omitted.
**(AR2.27) We will decrease the level of detail in this paragraph by removing the sentences describing the values for ice thickening rate, ice velocities and surface mass balance during this adjustment. We will also remove former Figure 3 and replace it with a figure showing the modelled ice velocities at 500 year intervals (Revised Figure 3).**
**(AE2.27) P12L29: We decreased the level of detail in this paragraph by removing the sentences describing the values for ice thickening rate, ice velocities and surface mass balance during this adjustment. We also removed former Figure 3 and replaced it with a figure showing the modelled ice velocities at 500 year intervals (Revised Figure 3).**

(RC2.28) P15L8: Typo "reconsutrction"
**(AR2.28) We will correct this in the revised manuscript.**
**(AE2.28) P16L9: The typo has been corrected.**

(RC2.29) P19L2-5: Check units!
**(AR2.29) This will be changed to mm (d K)$^{-1}$ both here and in Table 1.**
**(AE2.29) P21L8-L10 and Table 1: Units changed to mm (d K) $^{-1}$ for clarity.**

(RC2.30) Figure 9: Legends too small.
**(AR2.30) The legend size will be increased to 170% and moved out of the first panel**
**(AE2.30) Figure 10: The legend size has been increased to 170% and it has been moved out of the first panel**

(RC2.31) P23L25: than the => than that of the
**(AR2.31) This will be clarified.**

[revised manuscript text omitted]

---

## Author Response (AR2)

**Point-by-point replies and description of edits made to the manuscript in response to the reviewer comments by Alexander Robinson regarding manuscript "Simulating the Early Holocene demise of the Laurentide Ice Sheet with BISICLES (public trunk revision 3298)" by Ilkka S. O. Matero et al.**

https://www.geosci-model-dev-discuss.net/gmd-2019-304/#discussion

We thank both our reviewers again for their positive comments and Alex Robinson for raising the following comments. As highlighted by the reviewer, the description and discussion of adaptive mesh refinement in the manuscript can benefit from inclusion of further details and clarification. We have addressed the raised comments as discussed in our responses below. The reviewer comments and our responses are numbered and labelled respectively with (RC) and (AR).

**Point by point replies and description of edits done to the manuscript:**
(RC1): The manuscript as it is now is clearly improved over the original submission. We have improved the figures significantly and now help to explain the messages in the paper. The scientific quality is high and the experiments are interesting. There has also been a clear effort to show how the manuscript fits into the topics of GMD. I still feel some detail could be added to the technical aspects of the experiments, but overall, I would suggest publication with only minor changes.
**(AR1)**: Thank you for the positive comments.

(RC2): The discussion of the mesh refinement experiments needs improvement. Figure 7 does a good job now of highlighting the differences between the two resolution simulations (standard and AMR_2). And I believe the sentence on P19L1-5 summarizes the conclusion of this permutation well based on what is seen in the figures. But the later discussion of this topic in Section 5 is inconsistent, particularly the sentence at P26L26-28: "Increasing the level of AMR in our simulations results in a divergent response in the timing of the saddle collapse (fig. 6a), suggesting that further refinement of the mesh could result in the ice saddle reaching flotation earlier, with knock-on effects on the dynamics of the collapse." Therefore, I suggest revising the paragraph of P26L23-32 carefully.
**(AR2)**: We have revised the paragraph starting at P26L25, and shifted the focus to the lowest resolution being sufficient for the first-order evaluation of the meltwater pulse released from the ice saddle as stated earlier in P19L1-L5. We modified the sentence highlighted by the reviewer at P26L34-L35 by removing the discussion of the divergent response, but left the hypothesized importance of further refinement to sub-kilometer scale in the text in order to not understate its potential role in accelerating the dynamical part of the saddle collapse.

(RC3): Note that I do not see this lack of impact of very high-resolution modeling in this context as a negative point. To the contrary, Ω0 could already be considered unprecedented high resolution for simulating the Laurentide, and that should be emphasized in the text.
**(AR3)**: We have added text stating that the base resolution is already unprecedentedly high in simulating the LIS deglaciation to P26L25-L27.

(RC4): And, in fact, since there was little change between Ω0 and Ω2, it would have been neat to see how much computation you could save and still have viable results (eg running a simulation of Ω2 or even Ω3, but with the lowest order

resolution at 20km or 40km). If you could save on computation, that would allow you to run ensembles with more members and arguably make a stronger case for accurately assessing the uncertainty. Adding such a test at this point in the review is not necessary, but it certainly would add value.

**(AR4)**: As mentioned in response to one of the comments in the previous round of review (AR1.6), we do not currently have resources for running additional simulations. This is however an interesting question to include in the planning of potential future work with the model setup.

(RC5): Related to the above, but this is minor: I noticed the comments about runtimes did not seem fully consistent. Please correct this and consider using a common metric, either model years per day or model years per hour, but not both:
P8: $\Omega 2$ = 396 model years per day = 16,5 model years per hour
P8: $\Omega 3$ = less than 10 model years per day
P17: $\Omega 2$ = ~50 model years per hour
P17: $\Omega 3$ = 1 model year per day = 0,04 model years per hour
It is definitely interesting information for the reader to understand what kind of computational undertaking these experiments entail.

**(AR5):** The runtime numbers have been changed in the revised manuscript to be consistent and representative of the average run speed using a certain level of refinement of the adaptive mesh at both highlighted locations (P8L18-L20 & P17L16-L18). A consistent metric of model years per day is now used throughout the text.

(RC6): As a last point about the mesh refinement (I promise!), I would like to see a few sentences about how the topography was generated for the different resolutions, as this was not clear. Was the topography generated at the highest resolution, and then aggregated to the lower resolutions as needed, or vice versa? This has implications as to how much detail is conserved in the high resolution meshes.

**(AR6):** Topography is generated at the coarsest level of refinement and upscaled as part of the mesh refinement process using linear interpolation for the refined regions as necessary. This information has been added to P8L28-L31.

(RC7): In the tables, I do not understand the column "Reference" – does it refer to sections of the paper? This should be explained in the caption.

**(AR7)**: Thank you for pointing this out. The references point to sections of the text discussing the specific model parameter (Table 1) and simulation (Table 2) in question. We have added the missing information to the caption of each table.

[revised manuscript text omitted]